# Improving flood damage assessments in data scarce areas by retrieval of building characteristics through UAV image segmentation and machine learning – a case study of the 2019 floods in Southern Malawi

Lucas Wouters[1,2], Anaïs Couasnon[1], Marleen C. de Ruiter[1], Marc J.C. van den Homberg[2], Aklilu Teklesadik[2], Hans de Moel[1]

[1]Institute for Environmental studies (IVM), Vrije Universiteit Amsterdam, De Boelelaan 1087, 1081HV Amsterdam

[2]510, an initiative of the Netherlands Red Cross, Anna van Saksenlaan 50, 2593 HT Den Haag

*Correspondence to*: Lucas Wouters (wouters.lucas@outlook.com)

**Abstract.**

Reliable information on building stock and its vulnerability is important for understanding societal exposure to floods. Unfortunately, developing countries have less access to and availability of this information. Therefore, calculations for flood damage assessments have to use the scarce information available, often aggregated on a national or district level. This study aims to improve current assessments of flood damage by extracting individual building characteristics and estimate damage based on the buildings' vulnerability. We carry out an Object-Based Image Analysis (OBIA) of high-resolution (11 cm ground sample distance) Unmanned Aerial Vehicle (UAV) imagery to outline building footprints. We then use a Support Vector Machine Learning algorithm to classify the delineated buildings. We combine this information with local depth-damage curves to estimate the economic damages for three villages affected by the 2019 January river floods in the Southern Shire basin in Malawi and compare this to a conventional, pixel-based, approach using aggregated land use to denote exposure. The flood extent is obtained from satellite imagery (Sentinel-1), and corresponding water depths determined by combining this with elevation data. The results show that OBIA results in building footprints much closer to OpenStreetMap data, where the pixel-based approach tends to overestimate. Correspondingly, the estimated total damage from the OBIA is lower (€10,140) compared to the pixel-based approach (€15,782). A sensitivity analysis illustrates that uncertainty in the derived damage curves is larger than in the hazard or exposure data. This research highlights the potential for detailed and local damage assessments using UAV imagery to determine exposure and vulnerability in flood damage and risk assessments in data-poor regions.

## 1. Introduction

Worldwide, flooding is one of the most common and damaging natural hazards in both monetary terms and loss of life (UNDRR, 2019). Estimating flood damage is essential for shaping flood risk management before and disaster response after a flood. This can be done a-priori to support strategic risk reduction by, for example, increasing awareness in areas that are high in potential damage and therefore reduce vulnerability, or after a given flood event to quickly derive estimates of building damages to help with recovery and prioritize actions. This latter one is known as a Damage and Needs Assessment (DNA), which is usually based for the most part on data collected on the ground. For DNAs, household field surveys are conducted, as rapid DNAs and Post Disaster and

Needs Assessments (Jones, 2010). A-priori flood damage assessments are generally modelled and require extensive datasets on flood hazard characteristics, the exposed elements at risk, and the vulnerability of these elements (Budiyono et al., 2015; Alam, A. et al., 2018; UNDRR, 2019). Much work has focused on improving these damage estimates, quantifying the effect of different flood scenarios and its consequences (Murnane et al., 2017; Jongman et al., 2012; de Moel et al., 2015). Unfortunately, sufficient information on the exposure and vulnerability is often lacking or less accessible in developing countries (van den Homberg and Susha, 2018). Therefore, calculations for flood damage assessments must use the scarce data available, often aggregated on high national or district level. This lack of data complicates accurate and downscaled flood damage assessments, as shown in studies by Amirebrahimi et al. (2016) and Fekete (2012). The lower spatial level is, however, required for most flood risk management applications. Especially building damage remains hard to quantify, as existing classification categories often neglect spatial heterogeneity. This causes many uncertainties in the assessment about physical structure, content, and flood susceptibility (Wagenaar et al., 2016). Flood damage assessments are a standard procedure to identify potential economic losses in flood-prone areas. With growing populations and economies, the need to accurately estimate flood damage is gaining greater importance (Merz et al., 2010). Such assessments can enable the allocation of resources for recovery and reconstruction by humanitarian decision-makers when a disaster does strike (Díaz-Delgado and Gaytán Iniestra, 2014). For example, severe floods in January 2015 have demonstrated the need for improved flood damage assessments in Malawi. During this period, the worst flood disaster in terms of economic damage was recorded for 15 of its 28 districts, predominantly in the Southern Region. The total damage was estimated to be US$ 286.3 million, with the housing sector accounting for almost half of the total damage with US$ 136.4 million (Government of Malawi, 2015). More recently, the Chikwawa district was subjected to extensive flooding because of continuous rainfall by tropical cyclone Desmond in January 2019.

Several studies have suggested that flood damage assessments could be improved by incorporating the vulnerability of building structures. Blanco-Vogt et al. (2015) summarize different methods to retrieve building characteristics and estimate flood vulnerability based on building types in a semi-urban environment. Different building parameters are discussed that could affect the building susceptibility to flooding, including height, size, form, roof structure and the topological relation to neighbouring buildings and open space. Types are created by taking the remotely sensed data and relating this to potential flood impact. They note that these types can be used to link buildings to more detailed damage curves and discuss the challenges in terms of data resolution and techniques in remote sensing. The research of De Angeli et al. (2016) builds on the method of Blanco-Vogt et al. (2015) by developing a flood damage model that differentiates the urban area (using building clusters based on building taxonomies and footprints), instead of using a single homogenous land-use class. Remotely sensed data were used to derive exposure and vulnerability information after which it was combined with available building information. The model was able to assess damage estimates in an urban setting, with the total average damage deviating from the refund claims with a percentage error lower than 2%. Nonetheless, the authors state that a generalization of the procedure needs to be studied further. Another example of using indicator-based approaches regarding physical vulnerability, specifically tailored for data-scarce regions, is given by Malgwi et al. (2020). In this study, a conceptual framework is proposed that combines vulnerability indexes and regional damage grades (frequently observed damage patterns) by utilizing a synthetic 'what-if' assessment by experts.

Remote sensing has the potential to generate information on the exposure and vulnerability input for damage assessments. Numerous studies have been carried out for mapping land cover, such as built-up areas, with varying methods and spatial scales (Mallupattu and Sreenivasula Reddy, 2013; Ai et al., 2020). With new innovations in the resolution of imagery, also smaller-scale studies can be conducted where remote sensing can be applied to retrieve information on object-level (Klemas, 2015; Englhardt et al., 2019). In a review by De Ruiter et al. (2017)

it is stated that common flood vulnerability studies that use land-cover types could be improved by incorporating object-based approaches, for example by developing vulnerability curves for different wall-material types. A technique to derive useful information from remotely sensed image data is Object-Based Image Analysis (OBIA). OBIA has the potential to identify exposed elements and its characteristics accurately when incorporated into a flood damage assessment but there is little literature combining the methods. The process involves grouping pixels

into objects based on their spectral properties or external variables, after which they are combined into spatial units for image analysis such as image classification (Blaschke, 2010). Spectral properties to group these objects could, for example, be the mean value or standard deviation of spectral bands of the image. A conventional workflow to conduct an OBIA consists of two major steps: (1) segmentation and (2) feature extraction and classification. Literature demonstrates that the relationship between the objects under consideration and the spatial

resolution is critical for the accuracy of segmentation and the OBIA, improving with the emergence of higher resolution imagery (Blaschke, 2010; Belgiu and Drăguţ, 2014; Xu et al., 2019).

In this research, automated object recognition and classification from high-resolution images, based on an OBIA workflow, is used to delineate and characterize buildings in a flood damage assessment. This object-based

approach is applied to the 2019 January flood event for three villages in the Lower Shire basin in Malawi and compared to a conventional flood damage assessment based on disaggregated census data and homogenous land-use pixels (pixel-based approach). By doing so, this study aims to:

- create a framework to incorporate OBIA in flood damage assessments.
- Assess the added value of high-resolution UAV imagery in creating object-level exposure and
vulnerability data.
- Compare flood damage estimates between an object-based and conventional pixel-based approach.

In the next chapters we will introduce the case study area and the data, methods and results related to the pixel-based and object-based approach. In addition, a sensitivity analysis is performed to illustrate which components

of the risk assessment are most important when it comes to uncertainty in damage estimates.

## 2. Study Area

Malawi is a landlocked country in sub-Saharan Africa, bordered by Zambia to the Northwest, Tanzania to the Northeast, and Mozambique on the East, South, and West. The country is vulnerable to a range of natural hazards

including tropical storms, earthquakes, droughts, and floods. Especially floods affect many sectors from agriculture to sanitation, environment, and education. A major contributing factor to this risk is the variable and erratic rainfall, which often causes flooding in lower-lying areas after falling in the highlands. Between 1946 and

2013, floods accounted for 48% of the major disasters in Malawi. With a large rural population mostly relying on agriculture, these disasters have a large impact on the national economy and food security of the population (World Bank, 2015).


The Southern District of Chikwawa is one of the poorest and most flood-prone in the country. In addition to being exposed to flooding frequently, the district is characterized by a largely rural population and home to highly vulnerable communities in terms of economic diversification, employment opportunities and access to social services (Trogrlić et al., 2017). The Shire River is the largest river in the country and starts from lake Malawi flowing towards Chikwawa and into the low-lying Mozambique plain, as shown in Figure 1. In the district of Chikwawa, our study area, the river meets a large flood plain called the Elephants Marshes. This floodplain is characterized by stagnant flows, with the marsh varying in size depending on the flow of the river. When rainfall is high, large areas may be underwater.


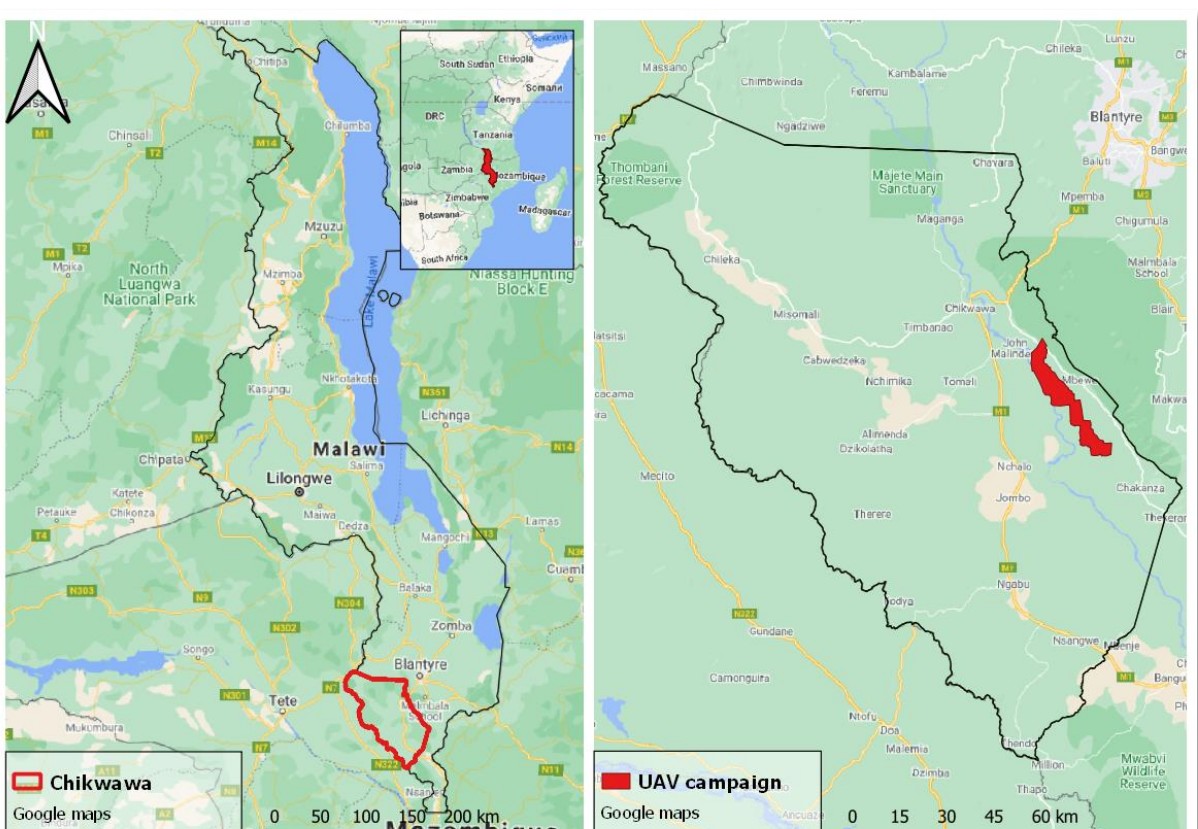

**Figure 1: The geographical location of Malawi (left) and the District of Chikwawa (right). © Google Maps, 2021.**

In 2015, Malawi underwent some of the worst flooding ever recorded in the country, affecting 1,101,364 people, displacing 230,000, and killing 106. In the aftermath, it became clear that the housing sector accounted for most of the total damage with almost 40% followed by agriculture with approximately 20%. The worst affected districts were in the Southern region, being the districts of Chikwawa and Nsjanje, and the disaster sparked a discussion about a more responsible policy towards this type of events. One of the lessons learnt from this event was that the lack of disaggregated (spatial) data and information management slowed down the disaster response and could eventually slow down recovery efforts as well (PDNA, 2015).

Between the 22nd and 26th of January 2019, the Chikwawa district was again subjected to extensive flooding because of continuous rainfall by Tropical Cyclone Desmond. There is no specific empirical damage data available for our case study area. However, the International Federation of Red Cross and Red Crescent Societies (IFRC) issued an Emergency Plan of Action (EPoA) after the floods. Based on preliminary assessment carried out by staff members and volunteers from the Village Civil Protection Committee (VCPC) and Malawi Red Cross Society (MRCS), one of the most affected is the Traditional Authority of Makhuwira with a total of 2,434 collapsed houses (IFRC, 2019). In Chikwawa, a total of 15,974 people were affected, 3,154 houses damaged or destroyed, and 5,078 people reported to be displaced across at least seven camps set up by communities and government. Most of the affected houses were semi-permanent buildings, which are also common in our study area (IFRC, 2019).

## 3. Materials

In order to determine the hazard, exposure and vulnerability for both the pixel- and object-based approach, a variety of data sources have been used. This section describes these data sources, including remote sensing and other geospatial data (including UAV imagery), local survey, regional building statistics and the datasets used for the construction of (local) damage curves.

### 3.1 Remote sensing data

UAV imagery was collected in November 2018 by The Netherlands Red Cross (NLRC) and the MRCS for mapping and flood simulation purposes in the Lower Shire Basin. A fixed-wing UAV (Smartplane Freya) with $0.3 m^2$ wing area, weighing around 1.5 kg, and a RICOH GR II camera was used to obtain the UAV imagery. The drone usually flew around 300 meters altitude, having a flight time of around 60 minutes per battery and with a sidelap and overlap of each 70%. The flights were carried out without Ground Control Points (GCP). van den Homberg et al. (2020) give a detailed description of the UAV model and data collection. Agisoft Photoscan and Metashape software was used to stitch the images of the optical imagery and extract a Digital Surface Model (DSM) from the stereophotogrammetry. The extent of the flight coverage is shown in Figure 1.

In addition to UAV imagery, other remote sensing data were acquired from open-source databases, including the Shuttle Radar Topography Mission (SRTM) digital elevation model (DEM) collected by NASA and the SAR Sentinel-1 imagery collected by Copernicus (Farr and Kobrick, 2000). The High-Resolution Settlement Layer (HRSL) provides an estimate of the settlement extent and population density and was developed by the Connectivity lab at Facebook in combination with the Centre for International Earth Science Information Network (CIESIN) by using computer vision techniques to qualify optical satellite data with a resolution of 0.5m (CIESIN, 2016). The OpenStreetMap (OSM) contains a features layer of manually delineated objects and was used for validation purposes (© OpenStreetMap contributors, 2019). Table 1 summarizes the various datasets.

**Table 1: Available datasets in this research. Abbreviations: Digital Elevation Model (DEM), Digital Surface Model (DSM), Ground Range Detected (GRD), Malawi Red Cross Society (MRCS), OpenStreetMap (OSM), Shuttle Radar Topography Mission (SRTM) Synthetic-aperture radar (SAR).**

| Datasets and platforms | Type | Resolution (horizontal) | Data repository | Acquisition | Used for |
|---|---|---|---|---|---|
| *Remote sensing* | | | | | |
| Space Shuttle | DEM | 30m | SRTM, Earth Explorer | Unknown | Flood hazard |
| Satellite | SAR | 23m | Sentinel-1, Copernicus | 24-01-2019 | Flood hazard |
| UAV | Optical | 0.11m | MRCS | 11-2018 | Exposure / vulnerability (object-based) |
| UAV | DSM | 0.25m | MRCS | 11-2018 | Exposure / vulnerability (object-based) |
| *Geospatial* | | | | | |
| HRSL | Land cover | 30m | CIESIN | 2016 | Exposure (pixel-based) |
| OSM | Vector | Object | OpenStreetMap | n/a | OBIA validation |

## 3.2 Building data

To gain information about the building stock present in the case study area, Teule et al. (2019) conducted a field survey in 4 villages in, or surrounding the Traditional Authority Makhuwira, (including Jana, Nyambala and Nyangu). The data was collected by randomly selecting buildings in the vicinity of these interviews. In total, 50 buildings were sampled and assumed to be representative buildings in estimating the different building characteristics present in the area. The OSM data layer reports on a total of around 1350 buildings in the villages selected for our analysis. Figure 2 shows an example of one the sample buildings. The survey collected characteristics of potential flood vulnerability parameters, including size, height, roof material, wall material, and inventory of the house. These parameters were selected based on key features that characterize building types in the region along with their spectral differences, making them easier to detect from remote imagery.

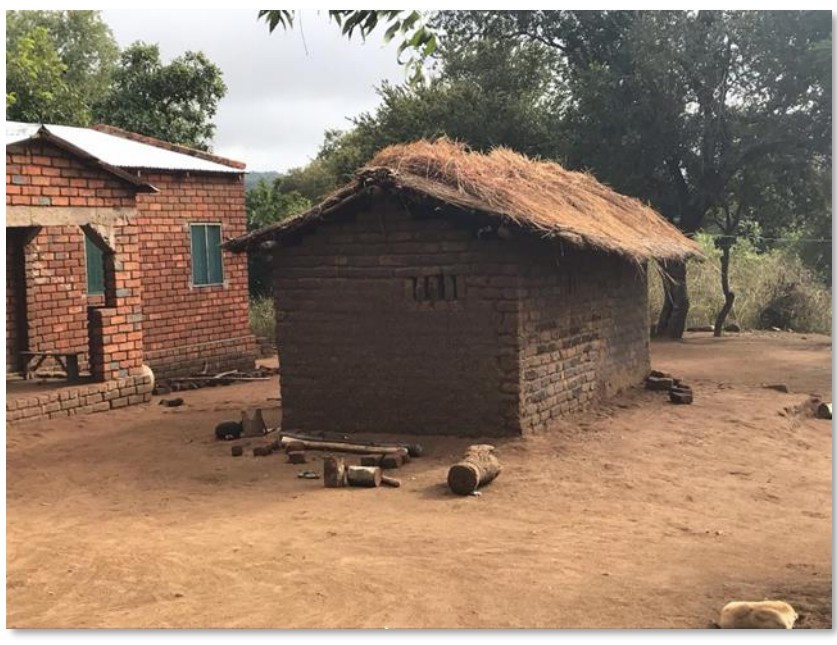

**Figure 2: Image from one of the sample buildings taken in the case study area taken by T. Teule (23-06-2019). A clear contrast between building material is visible between the two buildings: thatched roofs and unburnt bricks walls (middle building) versus iron sheeted roofs and burnt bricks walls (left building).**

In addition to the local field survey data, building stock used in the pixel-based approach was extracted from the Integrated Household Survey 2016-2017 (IHS4), conducted by the Malawi National Statistical Office (NSO) (Malawi Statistical Office, 2017). This report describes the distribution of three main building types by aggregating data to a regional level. A distinction is made based on their building material:

- A permanent building has a roof made of iron sheets, tiles, concrete or asbestos, and walls made of burnt bricks, concrete or stones.
- A semi-permanent building is a mix of permanent and traditional building materials and lacks the construction materials of a permanent building for walls or the roof. That is, it is built of non-permanent walls such as sun-dried bricks or non-permanent roofing materials such as thatch. Such a description would apply to a building made of red bricks and cement mortar but roofed with grass thatching.
- A traditional building is made from traditional housing construction materials such as mud walls, grass/thatching for roofs, or rough poles for roofbeams.

The ratio of the different dwellings in the district of Chikwawa is summarized in table 2. From this information, a trend can be observed towards a ratio with more formal buildings. The most recent statistics – being the building stock information from 2016-2017 – is used in the pixel-based flood damage assessment.

**Table 2: Ratios of building stock in the Chikwawa district of Southern Malawi (Malawi National Statistical Office, 2018).**

|  | Permanent (%) | Semi-permanent (%) | Traditional (%) |
|---|---|---|---|
| **Building stock 2010 - 2012** | 25.5 | 15 | 59.5 |
| **Building stock 2016 - 2017** | 33.7 | 33.8 | 32.5 |

### 3.3 Damage curve data

Stage-dependent damage curves are created for different building types by extracting material-specific vulnerability functions from the CAPRA platform. This platform contains a library with pre-defined analytical vulnerability functions, including different construction materials, calibrated with expert-supplied parameters (CAPRA, 2012). These curves express relative damage as a percentage with respect to water depth. Several examples in the library include concrete, wood, reed, masonry and earth (unfired) materials.

In addition to the vulnerability curves, maximum building damage values were estimated based on the different kind of materials and the costs of buildings found in Southern Malawi (Table 5). The values were validated by local authorities in the case study area during interviews by Teule et al. (2019).

### 4.  Methods

This section describes the two flood damage assessment methods compared: first, the conventional, pixel-based method, and second the proposed object-based method, after which their distinctive components are discussed in more detail.

### 4.1 Flood damage assessment

Figure 3 presents the workflow applied in this study to derive the flood damage estimates from the January 2019 flood event in the case study area. Following the general procedure of a flood damage assessment, both approaches can be divided into three separate components: hazard, exposure and vulnerability (Merz et al., 2010; de Moel and Aerts, 2011; Jongman et al., 2012), see Figure 3. In this research, we define the hazard as the flood extent and depth of a flood event, exposure as the exposed buildings to this flood and vulnerability as the susceptibility of these buildings to flooding.

For the **pixel-based approach**, the HRSL land-use map, containing homogenous land-use pixels, is used to determine the built-up area. Building stock information from Table 2 is used to create corresponding stage-damage curves for the defined building types (Malawi Statistical Office, 2017).

For the **object-based assessment**, we combine information from an OBIA of high-resolution UAV imagery with the stage-damage curves created from the field observations. Building footprints are detected and classified based on their aerial features to identify local building types. Local stage-damage curves are then assigned to these types by assessing the vulnerability of buildings found in the field survey.

Both flood damage assessments are inherently different on the classification of exposed elements and their flood susceptibility. In the terminology of the UNDRR (2019), this translates into different input data for the exposure and vulnerability components.

The two different approaches share the same hazard component, being the 2019 flood event. Based on Sentinel-1 satellite imagery, a flood extent is created, and its related water depth is estimated. The economic damage is

calculated by combining the flood impact with the different sets of exposure data and damage curves (Figure 3). In order to determine the relative influence of the different components on the resulting risk, we evaluate the influence of building size, water depth and damage curve on our damage assessment model using a one-at-a-time sensitivity analysis, as applied in Ke et al. (2012).

255

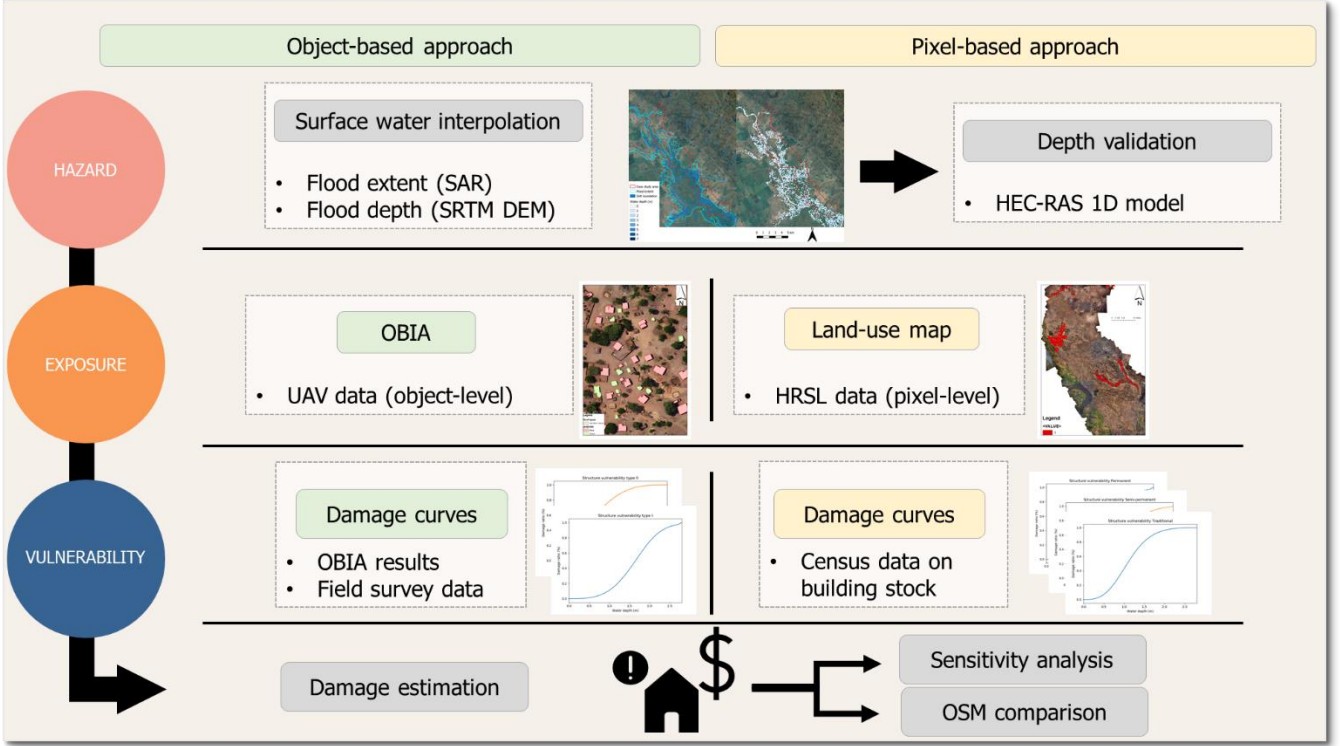

**Figure 3: Workflow of the two approaches of flood damage estimation. The left panel shows the object-based approach, the right panel shows the pixel-based approach. Abbreviations: Synthetic-aperture radar (SAR), Shuttle Radar Topography Mission (SRTM) digital elevation model (DEM), Object-Based Image Analysis (OBIA), High-Resolution Settlement Layer (HRSL), OpenStreetMap (OSM). The inundation (hazard) map is shown on © Google Satellite. The OBIA and land-use map are created using UAV imagery from the Malawi Red Cross Society.**

260

### 4.2 Hazard: flood area and water depth estimation

To represent the flood hazard, we derive water depths from the January 2019 flood event using the workflow presented in Figure 4. This approach takes the following three main steps: (1) extracting SAR data and processing it using SNAP software (SNAP, 2019) to create a flood extent map, (2) preparation of the data in ArcGIS and (3) using the available SRTM DEM to estimate the water surface elevation and extracting the flood water depth.

265

Extracting of SAR data and its processing was based on the SNAP flood mapping workflow (McVittie, 2019). This involved pre-processing of the SAR-imagery through calibration to transform the pixels from the digital values recorded by the satellite into backscatter coefficients, speckle filtering using the 'Lee filter' to remove thermal noise and geometric correction using the terrain correction function. Water and non-water are separated through setting a threshold by analyzing the backscatter coefficient histogram and manually determining the peak characteristics of land and water areas. Flooded areas could then be determined by setting a threshold value of 0.0022 which was defined based on the histogram plot of pixel values for reflectivity.

270

275

The flood raster map was further prepared in ArcGIS by vectoring the resulting water pixels using the 'Raster to vector' tool and aggregating with the 'Aggregate polygons' tool based on a neighborhood of 100 meters. Single-pixel polygons were removed to exclude noise from the flood map and any empty spaces in the polygon were filled using the 'Union' and 'Dissolve' tools. These filled spaces can be the result of beneath-vegetation flood areas that can be missed by the SAR processing (Shen et al., 2019). Negative values are removed in the next, final step if they are a result of actual topographic factors, such as local hills.

The final step in this approach follows the research of Cian et al. (2018) and Cohen et al. (2018), where the flood boundaries along the water surface are used to estimate the elevation of the water surface. The boundaries of the derived flood extent were turned into points with the 'raster to point' tool, after which the elevation values were extracted from the DEM. The water surface was then computed using the 'Inverse Distance Weighting (IDW)' tool. Essentially, this means that pixels inside of the flood extent get the elevation value of the closest elevation points along the boundary. The water depth can then be calculated by deducting the initial DEM values from the assigned water surface values. Figure 4 visualizes the workflow and the resulting output.

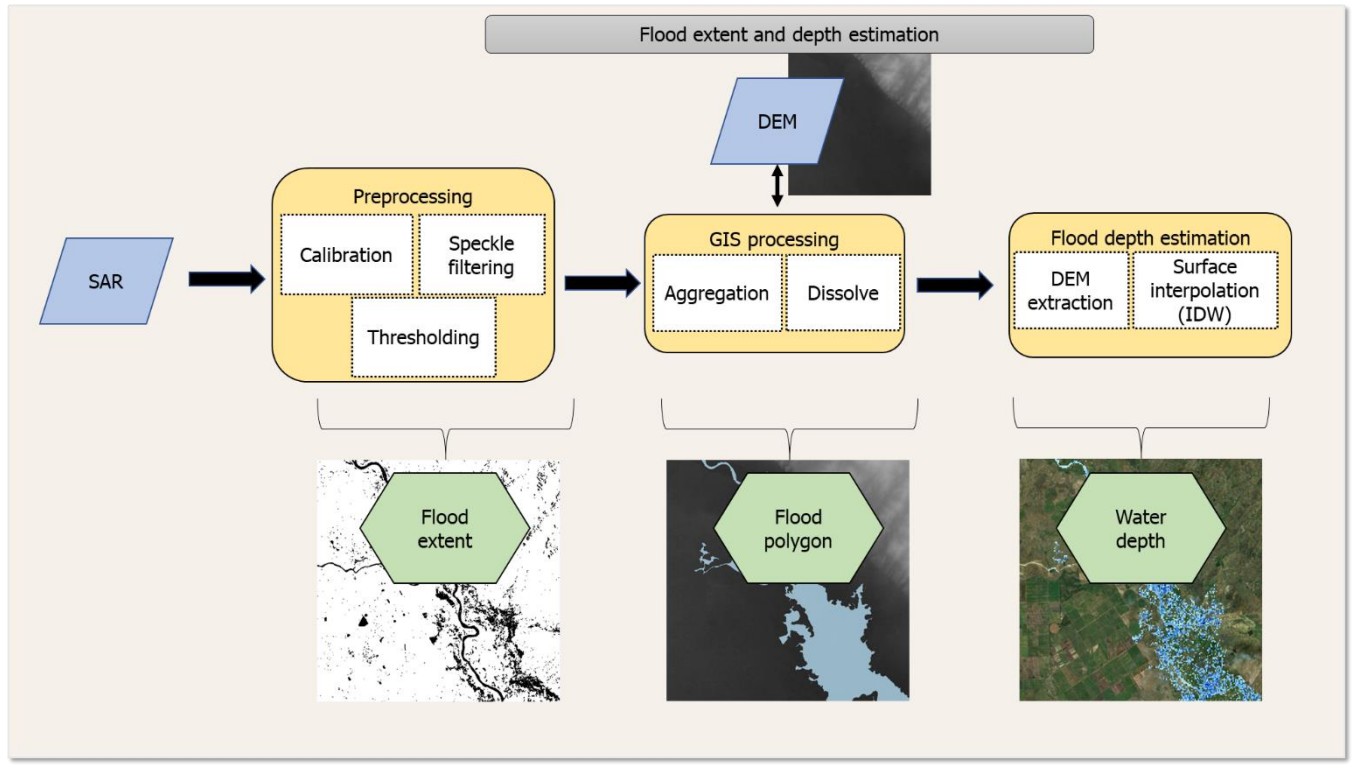

**Figure 4: The workflow representing the extraction SAR satellite imagery and deriving its corresponding water depth. The flood polygon is shown on SRTM DEM and the water depth map is shown on © Google Satellite.**

To validate the surface water interpolation method, the result is compared with a flood hazard map obtained from a hydraulic 1D steady model that was run for a subsection of the Shire river (Maparera River) in a study by Copier et al. (2019). The segment covers an area of 2.1 km$^2$ in which the river has a total length of 2.2 km. The model was run using Hydrologic Engineering Center's River Analysis System (HEC-RAS) software (Hydrologic Engineering Center, 1998). Due to a lack of historical data, the discharge values used as input for the model are

estimated to match the case study area's water flowing abilities without creating an extreme overflow. The discharge value was set to 50 m³/s and the Manning coefficient was set to 0.05. For both the surface water interpolation method and the hydraulic model run, the UAV DSM is used as input.

The Root Mean Square Error (RMSE) is used to evaluate the output from both approaches (Cohen et al., 2018). By doing so, it can be determined to what extent the output of both approaches deviate in terms of water depth estimation. In addition to the root mean, we also construct the Receiver Operating Characteristic (ROC) curve. The ROC curve is a probability curve and reports the True Positivity Rate (*TPR*), also called Recall (*R*), as a function of the False Positive Rate (*FPR*). The area under curve (AUC) represents the degree or measure of separability, with 1 representing a model with a perfect predictability, 0 complete unpredictability and 0.5 random guesses. The HEC-RAS flood map is taken as ground truth against the results of the surface water interpolation method, where a prediction can be either a True Positive (*TP*), False Positive (*FP*), True Negative (*TN*), or False Negative (*FN*). The *TPR* and the *FPR* are calculated as following:

$$TPR = \frac{TP}{TP + FN} \tag{1}$$

$$FPR = \frac{FP}{TN + FP} \tag{2}$$

### 4.3 Exposure

**Pixel-based approach**

For the pixel-based approach, the built-up area is estimated by taking the built-up area of the pixel according to average density percentages and building sizes. This density is determined by visual interpretation of the UAV imagery. The percentages found in distribution of building types reported by the IHS4 are used to calculate the damage corresponding to one pixel-unit.

**Object-based approach**

The OBIA consisted of the following steps (Figure 6). First, validation and training samples were collected from the villages in the case study area by manually delineating objects. We manually delineated a total of 144 building to serve as training and 556 as validation. This step was followed by segmenting the high-resolution imagery and classifying the vectorized objects. We selected the open-source geo-software Orfeo Toolbox (OTB). This toolbox is a library for image processing initiated by the CNES (French Space Agency) that includes numerous algorithms created for the purpose of segmentation and classification (Grizonnet et al., 2017).

Segmentation was performed using the Mean Shift Clustering algorithm utilized by OTB. The mean-shift algorithm exploited by Orfeo relates to the work of Michel et al. (2015), in which the goal of image segmentation is to partition large images into semantically meaningful regions. The following parameters were set: (1) the spatial radius or the neighborhood distance was set to 1.5m; (2) the range expressed in radiometry unit in the multispectral space to 5m; and (3) the minimum size of a segmented region to 5m², in relation to minimum building sizes. The Support Vector Machine (SVM) algorithm from the same Orfeo library served to classify the

vectorized objects from the segmentation. The SVM is a kernel-based machine learning algorithm that has been
effectively used to classify remotely sensed data (Mountrakis et al., 2011). The classifier was trained on samples
that represented the common features in the selected images and are summarized in Table 3. An example of the
output of this process is shown in Figure 5.

**Table 3: samples used as input for training the SVM classifier with mean value ranges of the spectral bands (nm).**

| Value | Label | Samples | Mean B0 | Mean B1 | Mean B2 |
|---|---|---|---|---|---|
| **1** | Vegetation | 28 | 121-164 | 135-165 | 101-136 |
| **2** | Metal | 27 | 207-241 | 207-244 | 205-245 |
| **3** | Thatch | 31 | 225-241 | 201-228 | 184-213 |
| **4** | Bare ground | 34 | 171-220 | 155-197 | 145-197 |
| **5** | Shadow | 24 | 113-154 | 114-150 | 113-137 |


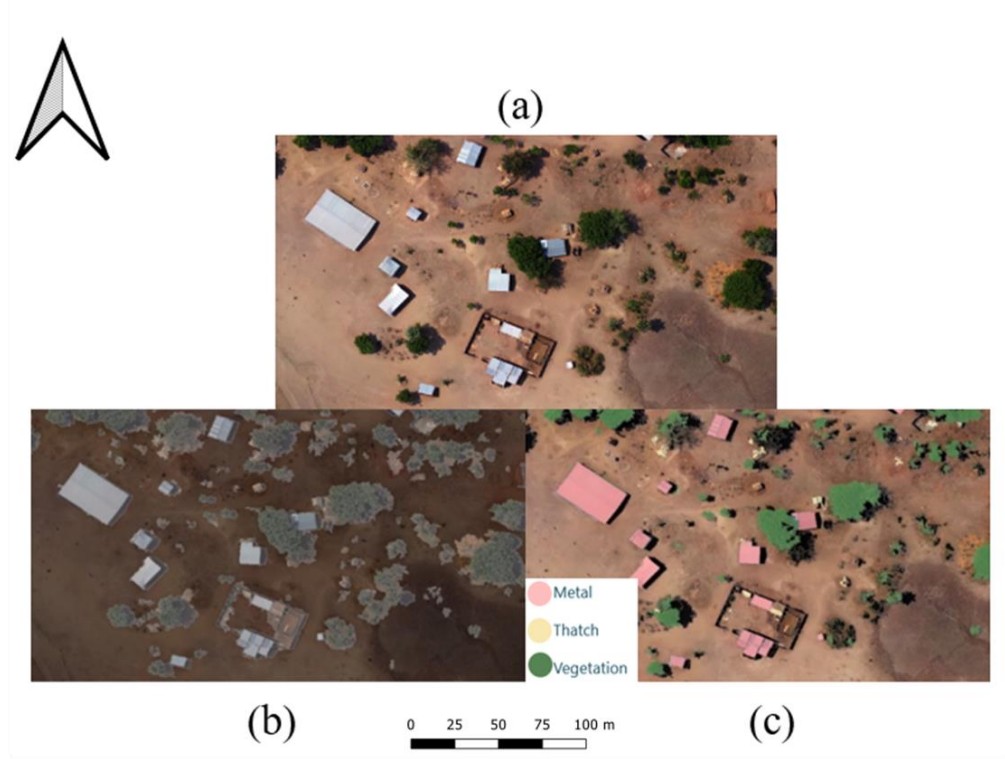

**Figure 5: Steps of the OBIA: (a) Original UAV imagery, (b) result of mean-shift segmentation, (c) classification using SVM classifier. The image contains UAV imagery collected by the Malawi Red Cross Society (MRCS), collected in November 2018. Made with QGIS.**

After the segmented objects were classified, a filtering process was conducted in which objects were removed
based on their respective height and category. By keeping the two categories that represent buildings with a height
over 0.5 m, buildings can be extracted, and potential misses are excluded from the damage calculation. This height
was chosen as a value between the height of the ground and a one-story building. The mean height from the DSM
was added to the objects by creating centric points of each segment and extracting the elevation values to these
points from the UAV DSM map. To derive the height of these objects, a baseline DEM was constructed and
subtracted from the mean DSM value. For this, the cells classified as 'Metal' and 'Thatch' were removed from

the DEM. Next, ground reference points were placed using visual interpretation to make sure no bushes or trees were selected. The elevation of these ground reference points was correspondingly used to interpolate an elevation surface using IDW (inverse distance weighting) and the elevation of this interpolated surface was used to

determine the height of the 'Metal' and 'Thatch' cells by determining the difference with the original DEM elevation.

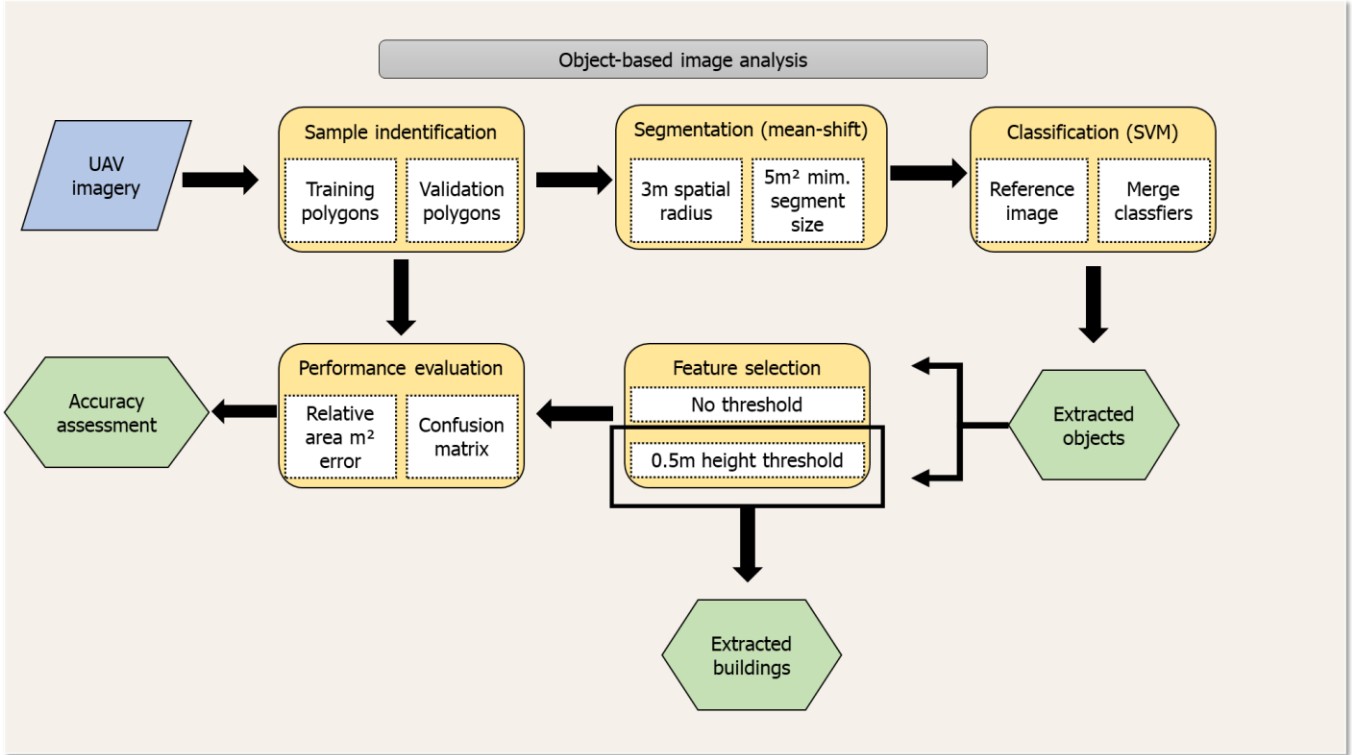

**Figure 6: Conceptual model of the building classification using automatic extraction methods.**

To evaluate the performance of the OBIA model, a map with 556 manually delineated/labelled reference buildings was compared to a map with predicted buildings from the classification. For this purpose, a confusion matrix was created (Gutierrez et al., 2020). In which, *TP* is the number of cases detected both manually and with the automatic approach. FP is the number of cases detected by the automatic approach but not manually. *TN* is the number of cases detected manually but not by the automatic approach. FN is the number of undetected cases. The statistical

parameters that were used to test the classification performance are the accuracy and F1-Score. The overall accuracy (*A*) was calculated given Eq. (1):

$$A = \frac{TP + TN}{TP + FP + TN + FN} \qquad \textbf{(3)}$$

To test the classification performance per class, the F1-Score was used. This statistic is the weighted mean of both

Precision (*P*) and *R*, where 0 indicated the lowest possible score and 1 a perfect score. The parameters are calculated with the following equations:

$$P = \frac{TP}{TP + FP} \tag{4}$$

$$F1 - Score = 2 * \frac{P * R}{P + R} \tag{5}$$

To evaluate the building area, predicted buildings were chosen that have partial or complete overlap with the reference buildings. From this selection, the Relative Error (RE) was calculated per building type. In this case, the absolute error is normalized by dividing it by the magnitude of the value of the reference buildings. The RE is calculated through the following expression:


$$RE = \frac{\sum_{n=1}^{N} |\theta^{\wedge} - \theta i|}{\sum_{i=1}^{N} |\theta i|} \tag{6}$$

Where $\theta^{\wedge}$ is the predicted value and $\theta i$ is the value of the reference buildings and N is the sample size.

## 4.4 Vulnerability: Damage curve estimation

Corresponding with the building types found in the exposure component of each flood damage assessment, a set
of damage curves is created. The description of the different types and their construction material is used to weigh material-specific damage curves from the CAPRA library, according to the method proposed by Rudari et al. (2016). We make use of the expanded aggregation table as proposed by Rudari et al. (2016), including the construction material considered for every building type (Table 4). This table indicates for each building type the building stock material, to which CAPRA damage curves are used.


For the pixel-based approach, three curves are created, for each of the building types (traditional, semi-permanent, permanent), making use of the description of wall building materials in the fourth Integrated Household Survey 2016-2017 (IHS4). Next, the distribution of these three building types (Table 2) is used as weights to create a single curve that can be applied to the urban pixels from the land-use map. For instance, semi-permanent housing
consists of unburned bricks, for which the Masonry and Earth CAPRA curves should be used. In this case, these curves are averaged and used to represent a semi-permanent building.

For the object-based approach, the results from the field survey are used to create damage curves for building types determined by aerial observation and the OBIA (Metal-roof and Thatch-roof). The materials of the roofs are
correlated to wall material, based on the field observations; from which we derive the wall-to-roof relationships. The local distribution found in wall material is used to weigh the curves from the CAPRA library based on percentages. This means, for example, that the distribution in wall material found for buildings with a thatch roof – being a burnt bricks, unburnt bricks, mud and wood – are used to weigh the CAPRA curves. These materials correspond to the Masonry, Masonry and Earth, Earth and Wood CAPRA curves, respectively.


In both approaches, we follow Maiti (2007) and assume that buildings constructed with a mud wall tend to collapse at a water depth of 1 meter. Before creating curves for each building type, this damage curve from the CAPRA library (Earth curve) is modified so that 1 meter of inundation corresponds to a 100% damage value.

**Table 4: Aggregation table of the CAPRA damage curves based on building stock information.**

| CAPRA materials | | | | Building stock material | Building types | | | | |
|---|---|---|---|---|---|---|---|---|---|
| Concrete | Masonry | Earth | Wood | | Metal-roof | Thantch-roof | Permanent | Semi-permanent | Traditional |
| | X | | | Burnt bricks | X | X | X | | |
| | X | X | | Unburnt bricks | | X | | X | |
| X | | | | Concrete | | | X | | |
| | | X | | Mud | | X | | | X |
| | | | X | Wood | | X | | | X |

## 4.5 Risk: Damage estimation

For the pixel-based approach, the built-up area is estimated by taking the built-up area of the pixel according to average density percentages and building sizes. The average density percentages and building sizes will be collected by visual interpretation of the UAV imagery. The percentages found in distribution of building types reported by the IHS4 are used to calculate the damage corresponding to one pixel-unit. The damage is calculated through the following expression:

$$D_{\mathrm{p}}[€] = \sum_{i=1}^{3} damage(i_{\mathrm{p}}) * a(i_{\mathrm{p}}) * r(i_{\mathrm{p}}) * rc(i_{\mathrm{p}})[€] \qquad (7)$$

Where:

- $i_{\mathrm{p}}$ = the building type (i.e. traditional, semi-permanent, permanent) as determined by the building stock description of the Malawi National Statistical Office (2018);
- $damage(i_{\mathrm{p}})$ is the damage per pixel in euros calculated with the adjusted stage-damage curve, and using as input the water depth [m] in the considered pixel;
- $a(i_{\mathrm{p}})$ is the size of the building in area m²;
- $r(i)$ is the ratio of the type according to the national survey in Table 2;
- $rc(i)$ is the replacement cost per m² based on the type ($i$), see Table 5.

For the object-based approach, damage is calculated per object by combining buildings automatically detected and classified through OBIA with the local stage-damage curves created from the field survey. The damage can be calculated through the following expression:

$$D_{\mathrm{o}}[€] = \sum_{i=1}^{2} damage(i_{\mathrm{o}}) * a(i_{\mathrm{o}}) * rc(i_{\mathrm{o}})[€] \qquad (8)$$

Where:

- $i_o$ = the building type based on the roof type and wall-to-roof relationships (i.e. metal-roof and thatch-roof);
- $damage(i_o)$ is the damage per building in euros calculated with the local stage-damage curve, and using as input the flood water depth [m] for this building.

### 4.6 Sensitivity analysis

To quantify how the damage parameters can influence the damage estimate, a one-at-a-time sensitivity analysis will be conducted by increasing and decreasing the different damage parameters with the mean of the respective relative errors. The sensitivity value (SV) will be used to represent the sensitivity and can be calculated by dividing the largest resulting damage by the smallest resulting damage (Koks et al., 2015).

## 5    Results

### 5.1 UAV imagery

Figure 7 shows the resulting UAV-based orthophoto, including the DSM with shaded relief, and the SRTM DEM with shaded relief, from the flight area. The Shire River is captured at the Western side of the acquired imagery. Completing the area took 140 flights, each lasting around 45 min. The UAV-based DSM shows a relatively equal elevation throughout most of the area. However, the absence of GCPs influenced the global accuracy of the elevation. A deviation can be observed when we compare the UAV-based DSM to the SRTM DEM. This DEM shows a down-sloping pattern of the elevation towards the south, in accordance with the flow of the Shire River.

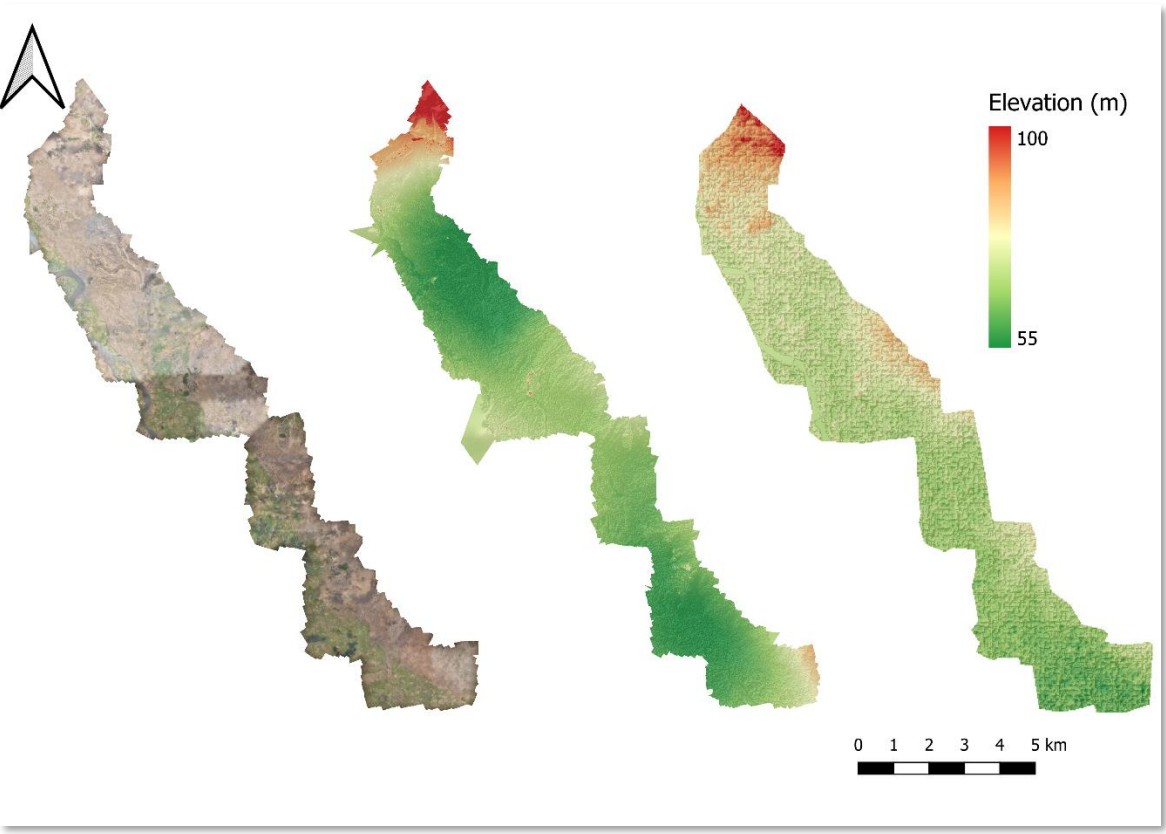

**Figure 7: Example of the orthophoto (left) and DSM middle) with shaded relief, produced with images from the UAV flight, and a Shuttle Radar Topography Mission DEM (right) with shaded relief. Made in Qgis using UAV imagery collected by the Malawi Red Cross Society (MRCS).**

Although this difference hampers large-scale (hydraulic) analysis using the UAV-based DSM, it is still valuable in assessing the local height variation of objects on a micro-scale. Figure 8 gives a detailed overview of the town of Chagambatuka, including the UAV-based DSM. Buildings can clearly be distinguished based on their rectangular shape and elevation, while trees are generally the largest objects in the area.


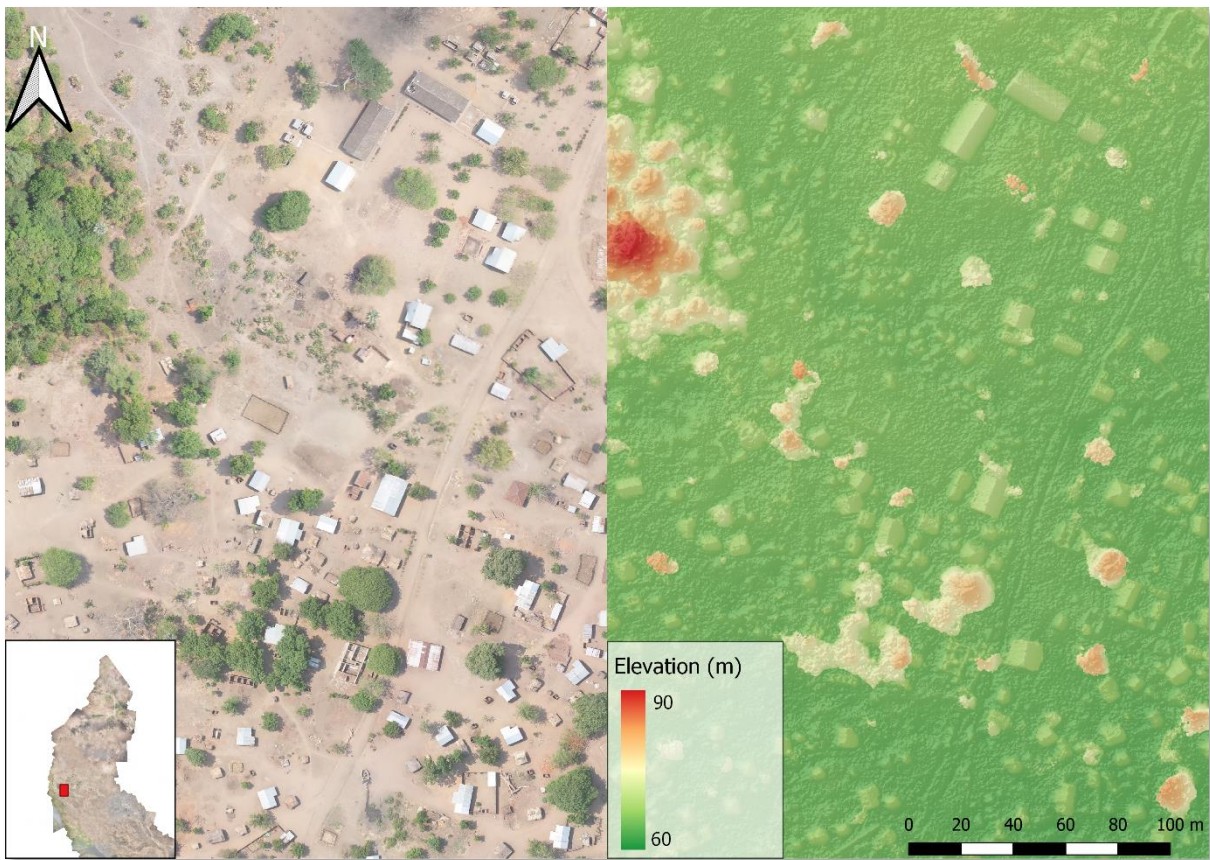

**Figure 8: Section of the town of Chagambatuka in the Northern part of the UAV campaign area, with orthophoto (left) and DSM with shaded relief (right), produced with images from the UAV flight. Made in Qgis using UAV imagery collected by the Malawi Red Cross Society (MRCS).**

**5.2 Field observations**

Based on the information collected through the building survey, buildings in the case study area are grouped into two types, based on their distinctive aerial features. From the 50 samples, no buildings were found that have a wall structuring resembling wood, reed or concrete. In addition, no buildings were found having tiles or any other material as the roof, nor any having more than two levels.


The first type of building in the area is composed of burnt and, in a small number of cases, unburnt bricks. This type is less vulnerable to flooding compared to the other type due to its material being less susceptibility to building failure. Its main distinctive aerial feature is a metal sheet roof, but the results of the OBIA and the field survey also indicate that this type of building has a larger footprint than thatch-roofed buildings. For the metal-

roofed buildings, two wall materials were found: burnt red bricks (90%) and unburnt bricks (10%).

The second type is generally composed of less formal building material, with its main distinctive feature being a thatch roof. The results of the survey seem to indicate a relatively equal distribution between the buildings materials, but as unburnt bricks and mud walls are more susceptible to building failure, this type is considered more vulnerable to flooding. For the thatch-roofed buildings, three wall materials were found: burnt red bricks (27%), unburnt bricks (41%) and mud/wattle (32%).

### 5.3 Damage curves and maximum damage values

Figure 9 shows the two damage curves created for the object-based approach based on types corresponding with the field survey (left) and three for the types in the building stock description of IHS4 national survey that are used in the pixel-based approach (right). Metal-roofed buildings show a lower vulnerability than thatch-roofed at the same water depth due to the structural integrity accompanied by more formal building material. Similar patterns can be observed for the damage curves based on the description of building stock in the Chikwawa district. Similar as to the damage curves have been derived, maximum damage values have also been determined. The damage values per square meters for all building types can be found in Table 5.

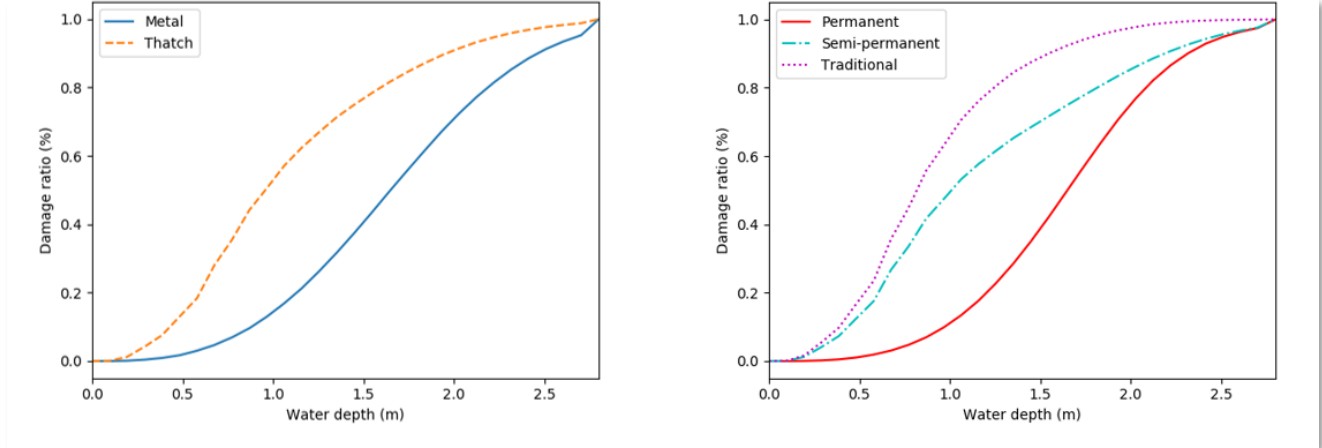

**Figure 9: Constructed damage curves for the two types derived from field and aerial observation for the object-based approach (left-hand panel), and three types derived from the description of building stock at district level for the pixel-based approach (right-hand panel) (Malawi National Statistical Office, 2017). The water depth is the flood water relative to the ground floor.**

**Table 5: Estimated maximum damage values per m² based on local knowledge of replacement costs (Teule et al., 2019).**

| Type | €/m² |
|---|---|
| Permanent | 15.20 |
| Semi-permanent | 10.60 |
| Traditional | 4.40 |
| Metal-roofed | 13.00 |
| Thatch-roofed | 9.70 |

### 5.4 OBIA quality assessment

The implementation of the OBIA model had a varying degree of success according to the statistical tests. Table 6 shows that classification is more reliable for classifiers that have a clear spectral difference with surrounding elements, such as shadow and metal roofs, whereas bare ground and thatched roofs are less easy to distinguish. This spectral difference resulted in a higher F1-Score for buildings with a metal roof (89%) compared to those with a thatched roof (53%). With the F1-score being the harmonic mean of the Precision and Recall, this metric

captures both the false negatives and the false positives of the classification process. The lower F1-score for detected thatch roofs could be attributed to their tendency to blend in with the environment because of their relatively similar spectral properties. With the addition of the height threshold for objects, the individual F1-scores for buildings were improved to 90% for metal-roofed buildings and 72% for thatched-roofed buildings. The increased F1-score for thatched-roof buildings indicates that having additional and accurate information on the

height of the objects has a large effect on the individual classification accuracy. The overall accuracy of the initial run shows a value of 77.45%, indicating the amount of correctly classified objects out of the total amount of samples. This value also increases up to 80% with the addition of a height threshold for objects, though this increase is also partly due to the exclusion of poorly performing classes such as 'Bare ground'.

**Table 6: Evaluation of the performance accuracy of the OBIA classification. *addition of height threshold by subtracting the extracted DSM and DEM values.**

| Label | F1-score | F1-score* | Accuracy (%) | Accuracy (%)* |
|---|---|---|---|---|
| **Vegetation** | 0.91 | - | | |
| **Metal** | 0.89 | 0.90 | | |
| **Thatch** | 0.53 | 0.72 | 77.45 | 80.19% |
| **Bare ground** | 0.49 | - | | |
| **Shadow** | 0.90 | - | | |

The building objects from the OBIA are a direct result of the segmentation process, and the relative error seems to reflect the same pattern as the classification process. This means that buildings with a thatch roof tend to be

harder to detect because the model groups pixels together that represent different objects, such as bare ground and the thatch roof. For both types, the relative error between observed and predicted building area can be observed in Figure 10. For the thatch roof buildings, 50% of the predictions are found with RE lower than 30%. For the metal-roofed buildings, this same percentage of predictions are found with a RE lower than 7.5%. Generally, metal-roofed buildings tend to be larger in size than thatch-roofed buildings, with a mean building size of 39 m²

and 21 m², respectively. For both types, the RE tends to decrease as building size increases. This seems to be in line with literature where it is stated that if objects get closer to the size of the available spatial resolution, errors are more likely to occur (Blaschke, 2010).

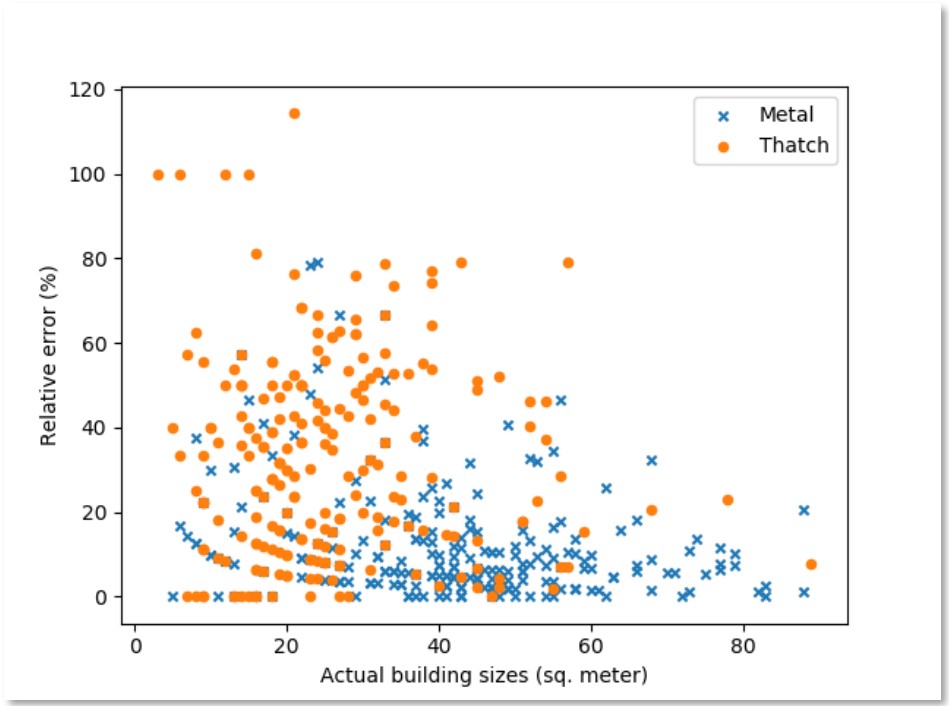

**Figure 10: Building area and relative error for both types (metal and thatch) in the case study area.**

## 5.5 Flood inundation

To check whether the surface interpolation method adequately captures flood characteristics, we compare our method with the results from a hydrodynamic model applied at the Maparera River. Figure 11 shows the maximum water depth obtained from both methods. The average water depth from the flood event at the Maparera river was 1.17 meter for the surface water interpolation and 1.22 meter for the hydraulic model run using the UAV DSM (Copier et al., 2019). The maximum estimated water depth for both approaches was about 3 meter (3.30 and 2.79 meters, respectively). The RMSE was calculated to be 0.73 meters. The results show that for a flood depth of approximately 3 meters, the surface water interpolation method deviated from the hydraulic model by <0.75 meters on average. We found considerable differences between both models along the main channel. This is in line with research from Cohen et al. (2018), given the inability of similar methods to calculate complex fluid dynamic effects. In addition, the interpolation method shows relatively low water depth at the upstream boundary compared to the hydraulic model. Nevertheless, the interpolation model seems to correctly dissolve the higher elevated area between the two main channels from the aggregated flood extent that was extracted from Sentinel-1 imagery. The AUC measured 0.73 (see Figure A1), indicating an acceptable agreement between the HEC-RAS reference map and the water depth map resulting from surface water interpolation (Hosmer and Lemeshow, 2000).

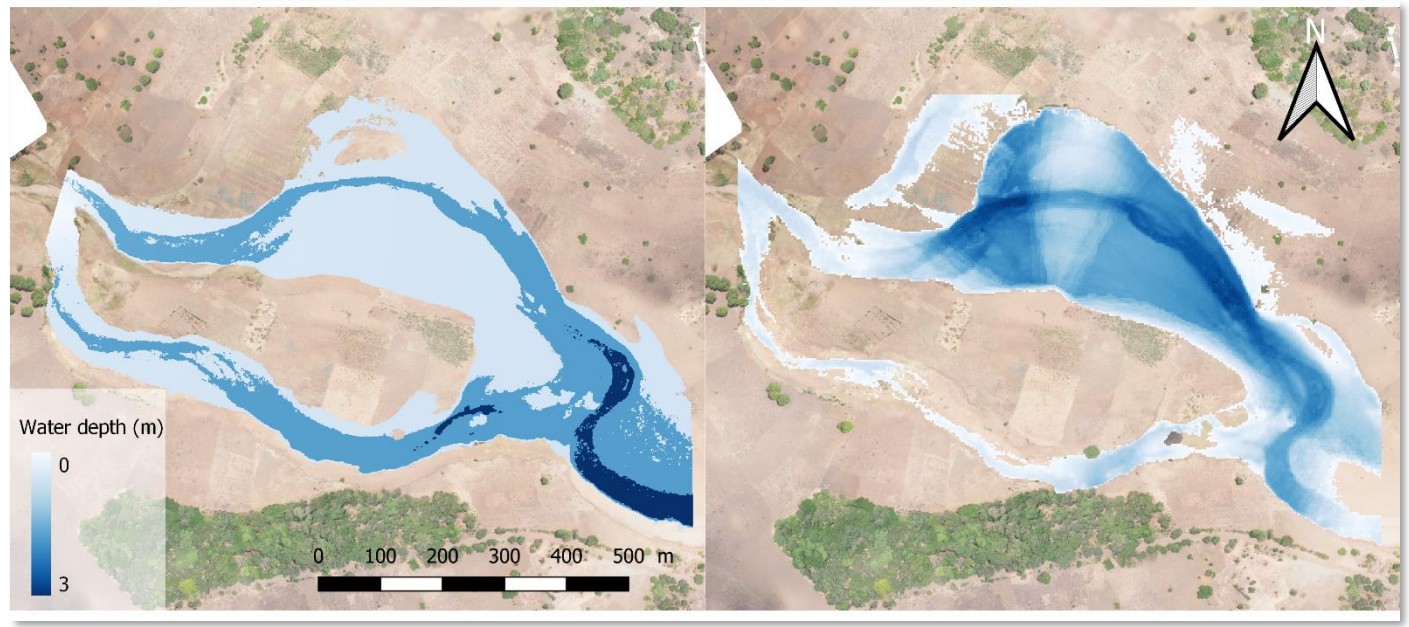

**Figure 11: The estimated water depth from the HEC-RAS hydraulic model (left) and the derived water depth following the surface interpolation method (right) at the Maparera River. Made in Qgis using UAV imagery collected by the Malawi Red Cross Society (MRCS).**


Repeating the method for the total case study area with the SRTM DEM produced a water depth map with an average water depth of 1.26 meter and a maximum water depth of 7 meters (see Figure 12). Buildings in the inundated area were assigned the water depth in the corresponding cell. Several areas with a positive water depth can be observed in the resulting flood map that deviate from the SAR inundation map. These areas indicate the

subtraction of incorrect water depth values from the DTM or capture areas that were not identified with SAR imagery, for example due to high vegetation.

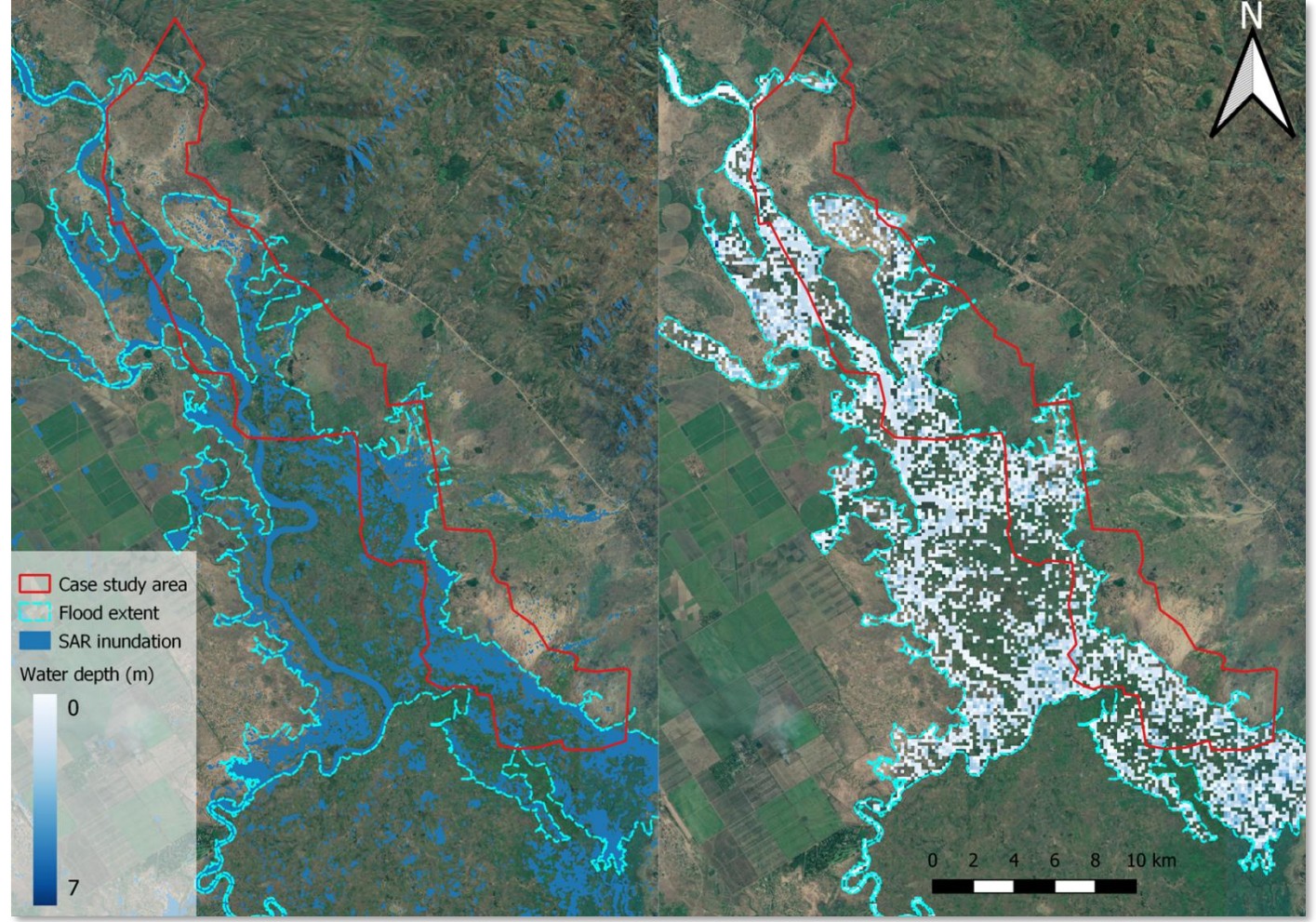

**Figure 12: The flood extent for the case study area extracted from SAR imagery (left) and the derived water depth map using surface water interpolation based on the SRTM DEM (right). The inundation maps are shown on © Google Satellite.**

### 5.6 Damage estimates

By overlaying the separate components of the flood damage assessment, the estimated damages were calculated for both approaches using equation 7 and 8. Compared to the pixel-based approach, the object-based approach provides a lower estimation of the exposed built-up area, of about two-thirds (Table 6). Interestingly, comparing the number of buildings in OSM, this comes very close to the amount extracted by the OBIA, giving confidence in the object-based approach. This amount of exposure influences the resulting damage considerably. The flooded built-up area for the pixel-based approach and the object-based approach was estimated at 2,541 m² and 3,952 m², respectively. This resulted in estimated flood damage of approximately €10K and €16K respectively (Table 6).

**Table 6: Flooded buildings and built-up area according to (1) the object-based approach, (2) pixel-based approach and (3) the available OSM map, and area and total damage according to (1) the object-based approach, (2) pixel-based approach.**

| Villages | Number of flooded buildings | | | Flooded built-up area (m²) | | | Total damage (€) | |
|---|---|---|---|---|---|---|---|---|
| | Object | Pixel | OSM | Object | Pixel | OSM | Object | Pixel |
| **1** | 9 | 11 | 10 | 371 | 338 | 348 | 1,286 | 1,754 |

| | | | | | | | |
|---|---|---|---|---|---|---|---|
| **2** | 54 | 92 | 61 | 1,424 | 2,768 | 1,321 | 6,215 | 10,043 |
| **3** | 21 | 28 | 26 | 746 | 846 | 732 | 2,639 | 3,931 |
| **Total** | 84 | 132 | 97 | 2,541 | 3,952 | 2,401 | 10,140 | 15,728 |

Although building densities and average buildings sizes were extracted from the same UAV imagery, a difference can be observed in the flooded built-up area between the two approaches. This is the result of the inability of land-use pixels to account for spatial variability of the buildings objects inside a certain area.

Similar research on flood events in urban and rural areas in Ethiopia, Germany and Poland, exemplifies that

significant uncertainties are present in flood damage assessments due to information lacking on the number of flooded buildings, the building types considered in the assessment and the distribution of building use within the flooded area (Merz et al., 2004; Englhardt et al., 2019; Nowak Da Costa et al., 2021).

**5.7 Sensitivity analysis**

By varying the building size and water depth parameters with the mean of the respective relative errors, the

sensitivity of the damage parameters for both approaches was estimated. As there is no information on the uncertainty of the damage curve values from the CAPRA database, the influence of this parameter is derived by using only the lowest and highest damage curve from the building types. For example, the lower damage bound for the damage curve sensitivity value in the object-based approach is computed by using only the Metal-roof damage curve and the higher bound using the Thatch-roof damage curve. Table 7 shows that the largest variance

in resulting damage is caused by this variance of the damage curves, meaning that the damage curve selection has the highest effect on the resulting damage estimates.

Table 7: The sensitivity values (SV) of the different damage parameters for the pixel- and object-based approach.

| | *Pixel-based* | *Object-based* |
|---|---|---|
| **Parameter** | **SV** | |
| **Building size** | 1.43 | 1.21 |
| **Water depth** | 1.46 | 1.56 |
| **Damage curve** | 1.71 | 1.90 |

Similar results have been found by Saint-Geours et al. (2015), in a cost-benefit analysis of a flood mitigation project where the uncertainty in the depth-damage curves is the prominent factor for the estimating of damage for private housing. Studies by de Moel et al. (2012) and Winter et al. (2018) also note that the most influential parameter in the uncertainty of flood damage estimates is the damage function. Moreover, it can be observed that the sensitivity value of building size is lower in the object-based approach compared to the pixel-based approach

(1.21–1.43), which can be attributed to less uncertainty in total building area that is flooded. This indicates that,

for the object-based approach, the increased accuracy with which buildings can be identified leads to a decrease in the uncertainty of damage estimates. The water depth parameter reveals that, although uncertainty in water depth results in varying damage estimates, sensitivity values for both approaches are comparable (1.46–1.56). Therefore, considering the same flood impact in each flood damage assessment does not affect damage estimates differently.

It is apparent from table 7 that all parameters involved in the flood damage estimation include an amount of uncertainty, and this propagates in the total estimated damage. As the flood map in both calculations remained equal, the differences can be attributed to the sensitivity of the damage parameters on the building types and damage curve parameters, or the exposure and vulnerability component, respectively.

## 6   Discussion

The preceding sections illustrate that by using OBIA, flood damage can be estimated on the object-level using UAV-derived imagery to detect buildings and classify them based on aerial features. This contributes to the literature in several ways. Complementing a study by Englhardt et al. (2019), that provides an impressive first glance at studies that use object-based data to classify buildings into vulnerability classes, our approach enables using this information to calculate damage on individual building level. Therefore, our method provides more certainty on the number of flooded buildings, their size and location. A more recent study by Malgwi et al. (2021) suggest that using data-driven approaches, such as multivariate damage models, could further improve estimates in data-scare regions compared to more expert-based approaches. However, this is not always feasible if the scarcity of empirical loss data hinders the implementation of multivariate models as it is the case in most developing countries. Our study indicates, however, that OBIA combined with local data can accurately estimate flood damage in an area where such data is absent, for example due to its remoteness.

Although this research has uncovered several important factors in the estimation of flood damage based on building detection, the issue deserves further additional research. First, the method was created for a specific case study area with little variation in building types. Building extraction is herein limited to the available building stock in the area, in this case resulting in two types. For urban areas, classification confusion might occur due to the heterogeneity of building types and structural properties. This complication could yield more uncertainties in assigning appropriate damage curves to buildings, especially as large discrepancies in potential flood damage exist between urban and rural areas in developing countries (Englhardt et al., 2019). Another distinction should be made between when studying areas with river-floods or flash-floods, as capturing the latter with Earth Observation data becomes a challenging task due to the low frequency of satellite imagery acquisition (e.g. SAR acquisitions) relative to the sudden happening of flash-floods (Mouratidis and Sarti, 2013).

Secondly, the filtering of objects using the DTM deserves further attention. Evidently, the addition of an object height threshold by using the DTM does indeed lead to a significant improvement in accuracy of the OBIA. In our study, the baseline DEM was constructed by manually setting reference points. The method was successful as brushes and trees that resembled bare ground could be avoided and the absence of ground control points during the UAV mission did not hamper the analysis. However, this method does rely on manual filtering, a process that

is hard to scale and prone to human error. Preferably, this method should be automatized. Several novel methods to extract bare earth surface can be considered in the future, including open-source filtering methodologies such as the CSF-plugin developed by CloudCompare. Zeybek and Şanlıoğlu (2018) discuss several filtering algorithms that can reach excellent accuracies using high-resolution point cloud data collected by UAV. Considering this approach in future research could enhance accuracy while simultaneously improving reproducibility and

scalability. Along with additional automatization, altering the OBIA workflow, by including the height threshold before image segmentation, could potentially improve classification results. Kamps et al. (2017) report that the classification accuracy of their OBIA to improve by 15-26% following this workflow but note that combining orthophotos with elevation data could potentially lead to the propagation of errors due to mismatches in datasets.

The third aspect refers to the additional field survey. The acquired buildings samples and their wall-to-roofs ratios provide insight into the relations between the local elements and the remotely sensed characteristics. However, a larger number of samples would be necessary to provide a statistically sound justification of the assumption on this relation. Obtaining field observations could become a difficult task if the method is scaled up, but a promising line of research could be the implementation of services like Mapillary or Google Street View for this purpose.

Combining the findings from this kind of research with field surveys can, therefore, complement the conventional methods by aggregating accurate estimates on building sizes, density, and characteristics. This would decrease the amount of uncertainty incorporated in potential scaled-up assessments. The HRSL provides an impressive first glance at exposed settlements and can be used as a base layer to project the distributions of building exposure and vulnerability found in this study. This method resembles the study of De Angeli et al. (2016), in which clusters

are created using representative buildings. In this case, field observations from drones and services like Mapillary can be combined to create representative villages or towns.

Finally, the other sources of uncertainty accompanied by the damage estimation need to be further studied. Although they do not directly relate to the results of the exposure estimation, the sensitivity analysis in this

research confirms that parameters such as floodwater characteristics, maximum damage values, and the applied damage curves have a significant effect on the total flood damage. The RMSE of about 75 cm and AUC of 0.73, illustrates that deriving water depths directly from a DEM can give results different from the hydraulic simulations. It should be noted that both methods have their disadvantages. In the hydraulic simulation, for instance, Copier et al. (2019) lacked specific discharge information for the event so an estimate had to be made

there that would resemble flood levels. The surface water interpolation method, on the other hand, lacks the dynamics of the hydraulic simulation, but is more specific to the event considered in this study as it is based on the observed extent. To validate the water depth estimation, the effects of using a coarser resolution SRTM DEM in surface water interpolation should be tested. Preferably, validation data from hydraulic models is used that corresponds to the flood event that is extracted from satellite imagery. This way, differences due to discharge

uncertainties are limited. Another way of validating flood events is through the collection of community-based data. By interviewing residents, ground-based observations can be collected in ungauged areas to serve as input for detailed catchment modelling and validating output (Starkey et al., 2017). Also, the aggregation of damage curves based on building material could yield uncertainties in the resulting flood vulnerability. For a more accurate

appropriation of the damage susceptibility, individual building types could be subjected to detailed survey studies that include historic flood events and damage with the corresponding building material.

## 7    Conclusions and outlook

The purpose of this research was to create a flood damage model based on the automated recognition of buildings and their characteristics through UAV image processing. By doing so, improvements on the exposure and vulnerability component of flood damage assessments were assessed and evaluated by comparing this new approach to a conventional one based on pixel-based information from a land-use raster. The two flood damage models were applied in a rural and flood-prone area in Southern Malawi, with a building stock consisting of mostly semi-permanent buildings.

In terms of direct damage considering the replacement costs of buildings in the study area, the flood damage based on homogenous land-use pixels is about 50% higher than the object-based approach (15,000 € versus 10,000 €, respectively). The calculation is found to be most sensitive to the damage curve used, with a sensitivity value (highest divided by lowest estimate) of 1.71 and 1.90, for the pixel-based and object-based approach respectively. However, uncertainty in building exposure still results in sensitivities of 1.43 for a pixel-based approach and 1.21 for an object-based approach. This illustrates that accurate information on exposure is essential in accurately estimating potential damage from flood events.

The effects of including high-resolution elevation information in the OBIA were examined by including a height threshold for classified objects. Individual F1-scores of the object-based classification were improved from 0.89 to 0.90 for metal-roofed buildings and 0.53 to 0.72 for thatch-roofed buildings. These results show that the integration of accurate elevation data can improve standard classification schemes based solely on spectral bands. The relative error on the area of the detected buildings tends to be lower for larger buildings and buildings with a clear spectral difference with the surrounding area. The water depth, derived by interpolating the surface water boundaries of a remotely sensed flood extent, deviated on average 0.73 meters from a hydraulic model for a maximum water depth of approximately 3 meters. This validation was conducted for a subset of the case study river using a high-resolution DSM. For the same area, the obtained AUC is 0.73.

Based on the results of this study we find that the primary utility of high-resolution UAV imagery in flood damage assessment is to spatially locate buildings in inundated areas and retrieve their characteristics by creating types in combination with local observations. These characteristics can be used to develop stage-damage curves that represent the local building stock instead of using aggregated information that implies homogeneous land cover for large regions. Furthermore, the number of buildings and their respective area and occupancy type can be derived to estimate flood damage more precisely. This improvement in data availability has the potential to aid humanitarian decision-makers in choosing appropriate policies with regard to flood protection or determining threshold levels for effective early-action measures in the case of flooding.

**Appendix A**

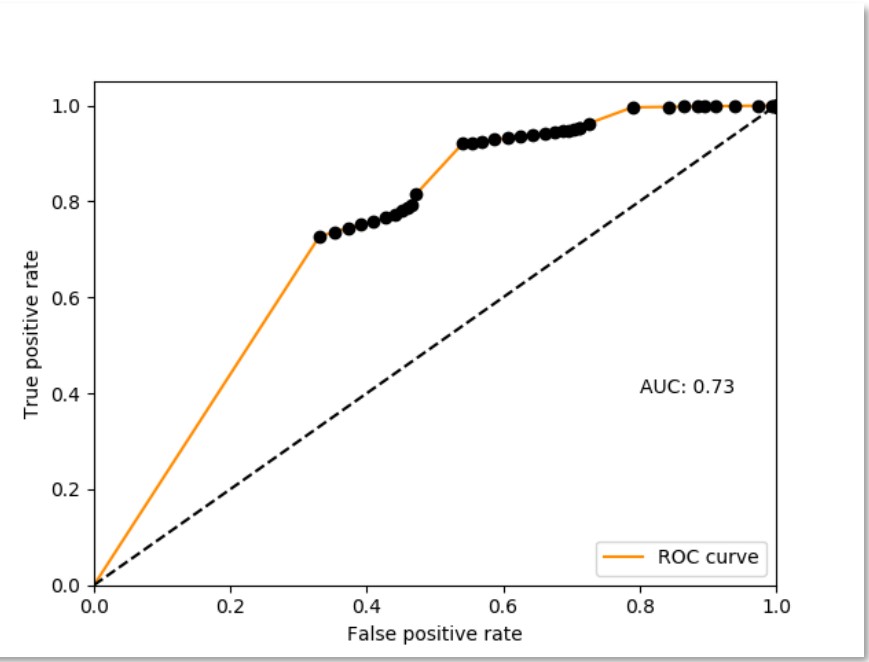

**Figure A1. The Receiver operating characteristic (ROC) and Area Under Curve (AUC), using the HEC-RAS as reference map against the water depth map derived from surface water interpolation.**

**Data availability.**

This work relied on data which are available upon request from the providers cited in Sects. 2 and 3.

**Supplement.**

-

**Author contributions.**

LW, HdM, MdR, AC and MvdH conceived the study. LW developed the theoretical framework and methodology with supervision from HdM, MdR, AC and MvdH. AK assisted in extracting and processing SAR data and helped LW carry out the inundation modelling. LW analyzed the data and prepared the draft, with all co-authors providing critical feedback and helping shape the analysis and manuscript.

**Competing interests.**

The authors declare that they have no conflict of interest.

**Acknowledgements.**

The authors would like to thank the Jurg Wilbrink (510) and Gumbi Gumbi and Simon Tembo from the Malawi Red Cross Society data team for obtaining the UAV data and Thirza Teule for collecting the essential field observations. The first and second European Civil Protection and Humanitarian Aid Operations (ECHO) program 725 on resilience building in Malawi financed the UAV mission. Anaïs Couasnon and Marleen de Ruiter acknowledge support from the Dutch Research Council (NWO) (VIDI; grant no. 016.161.324). We also thank the two anonymous reviewers for their valuable comments that helped improved the manuscript.

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
