# Peer review of "Improving flood damage assessments in data scarce areas by retrieval of building characteristics through UAV image segmentation and machine learning – a case study of the 2019 floods in Southern Malawi"

_Natural Hazards and Earth System Sciences, 2020_

## Referee Comment (RC1)

**Comment on nhess-2020-417**

The authors present an interesting study focusing on estimating flood damage in a study area in Malawi, Africa. Since this region can be considered data-scarce, a workflow has been established to better map and characterize buildings prone to flooding. The authors use UAV-borne imagery to extract single buildings using an object-based segmentation and classification approach. Based on the resulting semantic objects and their related vulnerability also assessed in the field a more detailed damage assessment is obtained. The results are compared to a more pragmatic approach based on available data.

To begin with, I have a remote sensing background and I'm not too familiar with assessing flood damage in practice. Please consider the comments with this regard. I have to admit that I was on the verge of rejecting the paper, due to methodological weaknesses. Working in a data-poor region can be somewhat challenging, but this argument cannot justify methods and results in conflict with scientific standards. However, I'd like to see if the manuscript can be improved after major revisions to meet the scientific standards of the journal. In the following you will find general and technical comments.

1) General comments

Paper structure

The structure of the paper could be improved by better differentiating between the applied workflows (OBIA vs. pixel-based). Further subsections on the study area and the UAV-results should be added. In the results section a third subsection level should be omitted.

Introduction

- Shorten the introduction to 1 page max
- Summarize the main objectives of the paper and what is the main innovation; ideally as bullet points

Materials and methods (instead of Data and methods)

- Add a subsection describing the study area in detail
  o Location, landscape, characteristics of the river, more details about floods in the past
- Please revise the workflow figure or add a second one to also include the main steps of the flood risk assessment and the main steps of the OBIA, the computation of the inundation maps and the modelling for validating the inundation maps
- Although presented in another paper, some basic information about the acquisition and processing of the UAV imagery is required since it is the primary dataset of the paper
  o Information about the campaign (planned acquisition, type of UAV + model and manufacturer, number of images, number of flight segments, selected overlap,…)
  o What are the regulations in Malawi regarding UAV flights above settlements? In European countries such campaigns would certainly be prohibited. Consider

mentioning this issue in the discussion section, this will have implications on the upscaling and transferability of the method
- o Software used (for planning and processing)
- o Results: also 3D point clouds?
- o GCPs used, georeferencing accuracy
- o Overview maps of UAV results
- Field survey: The cited EGU-Abstract does not include information about the recorded building characteristics. If there is no other source yet, it is encouraged to present the data in the results section.
- Assessment of object height (L210ff): This refers to the extraction of a digital terrain model. The state of the art for doing that is to use suitable filters applied to the derived 3D point cloud. This can also be done for point clouds derived from photogrammetric techniques based on UAV imagery, e.g. Zeybek and Sanlioglu 2019 (https://doi.org/10.1016/j.measurement.2018.10.013) and could be done easily e.g. using the CSF-plugin of the CloudCompare software. This would increase the objectivity and reproducibility of the results. Then the height of the objects could be inferred using "zonal statistics" instead of extracting the height at a single point.
- L237    Explain what you mean by "actual value". Does the actual error refer to the spatial correspondence of the reference compared to the classified objects?
- Consider changing the heading of section 2.3 to "Estimation of flooded area and water depth"; Parts of this methods section read like a step-by-step instruction; please revise
- L244f    Was the SAR data acquired at the peak flood (the greatest water depth)? If not, how does that affect the estimation of water depth?
- L248f    This is a procedure from the SNAP tutorial; add a citation for the tutorial document: http://step.esa.int/docs/tutorials/tutorial_s1floodmapping.pdf
- L270    Does the accuracy of the used data allow for extracting water depths of a few meters? According to the literature the vertical uncertainty of the SRTM data is several meters (e.g. Rodriguez et al. 2006, https://doi.org/10.14358/PERS.72.3.249). Both cited references use a far better baseline DTM (e.g. from LiDAR data) for this purpose.
- L274ff   The simulations of the hydraulic model used for the validation of the water depth should be described in more detail

Results

- Try to omit a third subsection level
- Add a subsection describing the resulting UAV data, including a map showing the UAV-based orthophoto and a shaded relief of the resulting digital surface model, including a detailed view
- Field observations: As far as I understand this subsection, the materials of the roof are correlated with the materials of the walls based on the 50 samples from field inspection, right? Please show the respective data, e.g. the percentage of cases for which these assumptions are valid. And is that really valid for the whole area then? From the outline of the study I expected that the main advantage of the UAV-campaign could be to automatically assess the wall materials and add them to the objects as an attribute.
- L357ff   Despite the even greater uncertainty of the used data, this is still a quite high deviation. Considering the created damage curves it makes a substantial difference, particularly for the thatch-/traditional type. Add a comment on that.

- Figure 6: The water depth map does not fully confirm the inundation area derived from SAR. There are significant areas with a positive water depth which do not appear in the SAR-based map. Please explain these deviations. This goes along with the previous comment in the M&M section, I guess the vertical accuracy may not allow to extract such shallow differences. For many areas you may end up with apparently negative water depths. Also the relatively coarse spatial resolution at least compared to the UAV-data may affect the actual water depth close to the buildings.
- L379ff   This is definitely the case. In a previous study it has been shown that including the normalized digital surface model (NDSM=DSM-DTM) can enhance the OBIA results substantiously from about 70% to 90% (Kamps et al. 2017, https://doi.org/10.3390/rs9080805). I wonder if this could apply in this study as well, particularly considering distinguishing thatched roofs from bare soil. By including the height above ground (NDSM) in the segmentation step (and not in the validation step) this should be possible.
- L407   can you estimate the damage down to the Euro or should it be rather rounded to the next hundred or thousand?
- L419   Is Merz et al. 2004 still state-of-the-art?
- Please explain the results presented in table 6 a bit more.

Discussion

- L444   Limited in terms of spatial coverage? Please explain.
- L449f   This is a good point, but it does not only apply to spaceborne SAR platforms. Also other types of satellite imagery (e.g. optical) could be used. Re-phrase the sentence accordingly.
- L451f   Do you refer to the materials of the walls and roofs here? Please explain.

Conclusions and outlook

Can you reflect on the ideal scale for establishing damage curves? Is it village per village or up to a whole province? How is such an optimal scale in line with the area which can be efficiently covered by a UAV campaign (order of a few squared km)?

2) Technical comments

**Abstract**

L17     estimate potential damage

**Introduction**

L30     The latter is known

L34     Alam et al.

L36     van den Homberg and Susha

L72     "smaller-scale" or "smaller-size"?

**M&M**

L112    OBIA is already defined earlier

L122    a flood damage assessment

L184f   Remove the last sentence

L190    of the neighbourhood distance

L191    minimum segment size to 5 m²

L192f   classify the resulting segments into semantic objects according to the desired classes

L205    the segments were classified into semantic objects

L202    greater than 0,5 m

L204    at these points

L219ff  These performance metrics are well-known, if you still decide to keep them in the text add respective citations; e.g. Roberts et al. 2019 (https://doi.org/10.3390/rs11161915), Gutierrez et al. 2020 (https://doi.org/10.5194/isprs-annals-V-2-2020-719-2020)

L236    typology -> type?

L241    where

L254    through setting

L256    histogram

L266    remove the S.

L310    that that

**Results**

L332    susceptible

L356    reference is missing

Discussion

L448    study areas

Conclusions and outlook

L480    structures structures

L500    potentially inundated areas (?)

Tables

Table 2 Bare -> Bare ground (?)

Table3  I'm not familiar with damage estimation for flood risk, but from other types of natural hazards the damage potential is often assessed based on the volume of buildings to account for multi-floor buildings as well. Consider adding a comment on that in the discussion section.

Table 4 Such classification results are typically too low to be accepted, please try to improve the classification as mentioned above.

Table 5 It seems like the other approaches may overestimate the damage compared to the UAV-based results. These results are thus more conservative which could be a better estimation in terms of help for recovery. If the actual damage would be underestimated due to systematic uncertainties handling the UAV-data, this would have consequences for the affected land/home owners. Consider this issue in the discussion section.

Figures

Figure 1        Improve readability, increase font size, mention UAV-data, hydraulic modelling is missing, SRTM-based inundation map is missing, comparison with OSM data is missing

Figure 2        Remove figure title and original caption, highlight the district in the left panel, add area of UAV-campaign

Figure 3        This photo is not part of the cited EGU-Abstract. Please cite the photographer and the date it was taken.

Figure 6        Revise figure caption – the left panel shows the SAR inundation, right? The SRTM data refers to the right panel.

Figure 7        How can the relative error be more than 100%? If the relative error is 100% the objects are not detected at all, right?

References

Please check the title of van den Homberg & Susha 2018: Combining UAV Imagery, Volunteered Geographic Information, and Field Survey Data to Improve Characterization of Rural Water Points in Malawi

---

## Author Comment (AC1)

**Authors' reply to referees' comments**

We would like to thank the reviewers for their very valuable feedback and detailed comments.

Following the suggestions of both reviewers, we will edit the manuscript at several places as described in detail in the individual comments. Mainly, the structure of the paper will be revised, additional information on the case study area and UAV data added, the method on creating damage curves clarified and more (recent) references discussed. Additionally, the technical comments will be addressed in the revised manuscript and figures will be edited to improve readability and include missing information.

Below, the general comments and questions will be addressed in italic. Any textual changes made in the manuscript are listed in blue.

We hope that our responses and the revised manuscript will meet your expectations. Thank you for your time and efforts, and for considering our manuscript for a possible publication in NHESS.

Yours Sincerely,

Lucas Wouters, on behalf of all co-authors.

**Referee #1**

**General:**

The structure of the paper could be improved by better differentiating between the applied workflows (OBIA vs. pixel-based). Further subsections on the study area and the UAV-results should be added. In the results section a third subsection level should be omitted.

**Authors' Reply (AR):** *Thank you for this suggestion and we agree with the reviewer that the manuscript benefits from an improved structure. To restructure the paper, we have decided to create similar subheadings in the Method (section 2) and Result sections (section 3). Furthermore, additional information on the study area and UAV-data has been provided in separate subsections. Finally, the third subsection level was removed.*

**Introduction:**

Shorten the introduction to 1 page max.

*(AR): The introduction was indeed rather lengthy and we have therefore, as suggested by the reviewer, shortened the introduction. However, we find that 1 page would not be sufficient for our multi-disciplinary study. As we combine different fields, we think it is important to provide the key concepts in both fields.*

Summarize the main objectives of the paper and what is the main innovation; ideally as bullet points.

*(AR): Based on the suggestion of the reviewer, the final paragraph of the introduction was restructured into bullet points to make the key novel points of our study more explicit.*

**Materials and methods:**

Materials and methods (instead of Data and methods)

*(AR): As suggested by the reviewer, the section has been renamed to Materials and methods.*

Add a subsection describing the study area in detail: location, landscape, characteristics of the river, more details about floods in the past.

*(AR): Additional information on the case study area and historic floods will be added to the revised manuscript.*

Please revise the workflow figure or add a second one to also include the main steps of the flood risk assessment and the main steps of the OBIA, the computation of the inundation maps and the modelling for validating the inundation maps.

*(AR): Based on this comment, we added individual workflow figures for the main steps of the OBIA, the computation of the inundation maps and the modelling for validating the inundation maps in the supplementary data section. Figure 1 visualizes the different components of a flood damage assessment, including the steps that were taken to estimate total damage. Flood risk, in this research, is defined as the combination of the physical characteristics of the flood event (the hazard) and its potential consequences (the vulnerability and exposure), whereas the flood damage is the monetized risk for a given flood scenario (the event discussed in our paper).*

Although presented in another paper, some basic information about the acquisition and processing of the UAV imagery is required since it is the primary dataset of the paper.

*(AR): Additional information on the collected UAV imagery will be added in the revised manuscript, including information on the flight mission and processing. Agisoft Photoscan and Metashape was used to stitch the images of the optical imagery (11 cm resolution). Also, this software was used to extract the DEM/DSM from the stereophotogrammetry, whereby we had a minimum of 70% sidelap and 70% overlap.*

Field survey: The cited EGU-Abstract does not include information about the recorded building characteristics. If there is no other source yet, it is encouraged to present the data in the results section.

*(AR): Following the suggestion of the reviewer (both reviewer #1 and #2), the results of the survey will be added to the results section.*

Assessment of object height (L210ff): This refers to the extraction of a digital terrain model. The state of the art for doing that is to use suitable filters applied to the derived 3D point cloud. This can also be done for point clouds derived from photogrammetric techniques based on UAV imagery, e.g. Zeybek and Sanlioglu 2019 (https://doi.org/10.1016/j.measurement.2018.10.013) and could be done easily e.g. using the CSF-plugin of the CloudCompare software. This would increase the objectivity and reproducibility of the results. Then the height of the objects could be inferred using "zonal statistics" instead of extracting the height at a single point.

*(AR): We thank the reviewer for this suggestion. This is a very interesting method which could certainly improve the reproducibility of the results, while at the same time, it could improve scalability of the method, as no visual interpretation would be necessary (if accurate enough). A paragraph on this technique will be added to the discussion section. However, we received the data in its current form and in the Zeybek and Sanlioglu (2019) paper it is also mentioned that ground control points are used to optimize accuracy. Unfortunately, our UAV lacks these observations, which could hamper this approach. Therefore, at this point it is not feasible to re-run the whole calculation considering the dataset we used as a starting point and the scale of the assessment. The revised manuscript will*

*however include a discussion of such novel methods and encourage future research to look at the impact of the selected technique used to derive the DTM.*

L237 Explain what you mean by "actual value". Does the actual error refer to the spatial correspondence of the reference compared to the classified objects?

*(AR): In this respect, the actual value refers to the size of buildings that were manually delineated through visual interpretation, also called reference buildings. The absolute error refers to the difference in buildings size of the predicted buildings – being either larger or smaller – compared to the reference buildings. This will be clarified in the manuscript.*

L274ff The simulations of the hydraulic model used for the validation of the water depth should be described in more detail.

*(AR): In the revised manuscript, more information on the hydraulic model will be added including model parameters and input data.*

**Results:**

Try to omit a third subsection level:

*(AR): A third subsection was dropped by renaming section 3.1.*

Add a subsection describing the resulting UAV data, including a map showing the UAV-based orthophoto and a shaded relief of the resulting digital surface model, including a detailed view.

*(AR): In the revised manuscript, the UAV data will be discussed in more detail including several figures of the results.*

Field observations: As far as I understand this subsection, the materials of the roof are correlated with the materials of the walls based on the 50 samples from field inspection, right? Please show the respective data, e.g. the percentage of cases for which these assumptions are valid. And is that really valid for the whole area then? From the outline of the study I expected that the main advantage of the UAV-campaign could be to automatically assess the wall materials and add them to the objects as an attribute.

*(AR): Correct, for the object-based approach, our local survey data (consisting of 50 buildings) is used to construct two building types based on their distinctive aerial features, being metal and thatch. For the metal-roofed buildings, two wall materials were found: burnt red bricks (90%) and unburnt bricks (10%). For the thatch-roofed buildings, three wall materials were found: burnt red bricks (27%), unburnt bricks (41%) and mud/wattle (32%). These percentages were used to weigh the material-specific curves from the CAPRA library and create a damage curve for both metal and thatch buildings. Ideally, a larger sample size would have been acquired to provide a statistically sound justification for using this data to create damage curves. Unfortunately, a narrow time window and limited means of communication and accessibility meant that this sample size was all that could be achieved. In the discussion, we refer to using street view imagery to not only derive the distribution of these wall-to-roof relations with greater ease and in larger sample size, but also pinpoint the location of these walls and combine them with remotely sensed data. Based on these recommendations, a tool is currently being developed that automatically detects and locates wall material from Mapillary street view (Gortzak et al., 2021). Although the Mapillary data used for the study of Gortzak et al., (2021) is based on an area in the North (out of our study area), it does show the potential of certain approaches.*

*Gortzak, I., van den Homberg, M., Margutti, J., Beddow, C., and van Aalst, M.: Characterizing housing stock vulnerability to floods by combining UAV, Mapillary and survey data – A case study for Karonga, Malawi, EGU General Assembly 2021, online, 19–30 Apr 2021, EGU21-12810, https://doi.org/10.5194/egusphere-egu21-12810, 2021.*

L357ff Despite the even greater uncertainty of the used data, this is still a quite high deviation. Considering the created damage curves it makes a substantial difference, particularly for the thatch-/traditional type. Add a comment on that.

(AR): *The RMSE of about 75cm illustrates that deriving water depths directly from a DEM can give results different from the hydraulic simulations. It should be noted that both methods have their disadvantages. In the hydraulic simulation, for instance, we lacked specific discharge information for the event so an estimate had to be made there that would resemble flood levels. The DEM method, on the other hand, lacks the dynamics of the hydraulic simulation, but is more specific to the event considered in this study as it is based on the observed extent. With adding the description of the hydraulic modelling, we will also reflect on those aspects in this section to give some context on the RMSE. Furthermore, we will add that a single flood hazard map is used in the rest of the study, meaning that this uncertainty would not impact our findings.*

*In addition, using the surface water interpolation we found considerable differences with the hydraulic model along the main channel. This is in line with research from Cohen et al., (2018), given the inability of similar methods to calculate fluid dynamic effects. Most buildings, however, are in the floodplain, further away from the main channel where large differences between this method and the hydraulic model occur. This means that a larger RMSE, caused by these differences, is not necessarily a reason to abandon our proposed approach. We do think that using the surface water interpolation method is a computationally simple method, using only imagery of a floodplain and a DEM. In addition, it does not rely on observed discharge values, contrary to hydraulic modelling, which makes it functional in data-scarce areas.*

Figure 6: The water depth map does not fully confirm the inundation area derived from SAR. There are significant areas with a positive water depth which do not appear in the SAR-based map. Please explain these deviations. This goes along with the previous comment in the M&M section, I guess the vertical accuracy may not allow to extract such shallow differences. For many areas you may end up with apparently negative water depths. Also the relatively coarse spatial resolution at least compared to the UAV-data may affect the actual water depth close to the buildings.

(AR): *The vertical resolution (full meters) is indeed an important limitation of using the global DEM in studies like these. Given that water depths are in the order of magnitude of a few meters, this can cause deviations in the resulting flood depth map indeed. We observe in our study, as in other global studies, that using coarse (both horizontal and vertical) resolutions usually result in overestimations of flood extents. To clarify this, we will discuss the areas that coincide and differ from the SAR image along with the above context in the revised manuscript.*

*As mentioned above, we will add an explanation of the hydraulic modelling and the influence of the DTM on the obtained water depth map. Indeed, as pointed out by the reviewer, those negative water depths could results from discrepancies between the inundation area derived from SAR and the DTM. For this study, we simply decided to ignore negative water depth as it was outside the scope of our study, but we recommend future studies to look at the influence of the DTM on the obtained flooded area.*

L379ff This is definitively the case. In a previous study it has been shown that including the normalized digital surface model (NDSM=DSM-DTM) can enhance the OBIA results substantiously from about 70% to 90% (Kamps et al. 2017, https://doi.org/10.3390/rs9080805). I wonder if this could apply in this study as well, particularly considering distinguishing thatched roofs from bare soil. By including the height above ground (NDSM) in the segmentation step (and not in the validation step) this should be possible.

*(AR): we fully agree with the reviewer that this is a very good idea and our results (Table 4) show the clear added value of incorporating this in an OBIA workflow. We will add it to the recommendation for future studies.*

L407 can you estimate the damage down to the Euro or should it be rather rounded to the next hundred or thousand?

*(AR): We agree with the reviewer that a modelling set up like this cannot estimate damages down to individual Euros. However, given that the objective is to compare the different approaches (Object, Pixel, OSM) and the model is deterministic in nature, the quoted damage numbers are mainly used to show the differences. Rounding (to the hundred or thousand level) would adjust these differences in an unwanted way. We therefore decided to leave the full numbers in the table, but adjust the text to the nearest thousand (i.e. approximately 10k and 16k in line 407).*

L419 Is Merz et al. 2004 still state-of-the-art?

*(AR): Much state-of-the-art methods are based on the framework of Merz et al. 2004. In the revised manuscript, more recent references will be cited that refer to these findings and we will rephrase the sentence to clarify this.*

Please explain the results presented in table 6 a bit more

*(AR): The sensitivity value presented in table 6 represent the ratio of the largest resulting damage divided by the smallest resulting damage. Therefore, it is an indicator of the spread of the uncertainty within our analysis. Therefore, a higher value means a larger uncertainty or influence of this parameter on the estimated damage. We will clarify this in the revised manuscript and refer to the numbers in table 6 to make the link more explicit.*

**Discussion:**

L444 Limited in terms of spatial coverage? Please explain.

*(AR): In this respect, we are referring to the distribution of building types in the area. In this study, only two types of roofing could be identified. This could limit the transferability of the method to areas with different roof types. This aspect will be explained more thoroughly in the discussion.*

L449f This is a good point, but it does not only apply to spaceborne SAR platforms. Also other types of satellite imagery (e.g. optical) could be used. Re-phrase the sentence accordingly.

*(AR): Thank you for pointing this out. We agree and the sentence will be rephrased in the revised manuscript.*

L451f Do you refer to the materials of the walls and roofs here? Please explain.

*(AR): Correct. The wall-to-roofs ratios found in the field survey provide the baseline for creating building-type specific damage curves. However, in an ideal situation, we would opt for larger sample size. This point will be clarified.*

**Conclusions and outlook:**

Can you reflect on the ideal scale for establishing damage curves? Is it village per village or up to a whole province? How is such an optimal scale in line with the area which can be efficiently covered by a UAV campaign (order of a few squared km)?

*(AR): Ideally, damage curves are created for each individual building by first detecting them from remote sensing imagery and then linking them to wall material using information from street view imagery. However, we understand that this is not feasible for every village considering accessibility and the effective range of drones.*

*Province level would be too large, considering the significant differences in building stock between, for example, rural and urban areas. Village level would be too local, as this might not be representative for a larger region. The sweet spot would be somewhere between these two levels, creating meaningful cluster of representative villages and by differentiating between rural and urban area. Preferably, several villages of different sizes are covered in one flight mission to create damage curves. This way, the area can practically be covered by drones, where the information can still be used for surrounding traditional authorities provided that they share similar geography. Following this rationale, one flight mission could create damage curves for around 3-5 surrounding traditional authorities, although this assumption deserves attention in further studies. To put this into perspective, Malawi has 28 districts and approximately 250 traditional authorities. The district of Chikwawa has 11 traditional authorities. However, not all traditional authorities are exposed to flooding.*

*Another division that can be considered, is a delineation based on meaningful geons. A geon is a region that is delineated based on uniform response to a phenomenon under space-related concern (Lang et al., 2014).*

Lang, S., Kienberger, S., Tiede, D., Hagenlocher, M., & Pernkopf, L. (2014). Geons-domainspecific regionalization of space. Cartography and Geographic Information Science, 41 (3), 214–226.

**Technical:**

*As mentioned above, most technical comments will be addressed in the revised manuscript. Specific comments that deserve a direct answer are noted here.*

Table 3 I'm not familiar with damage estimation for flood risk, but from other types of natural hazards the damage potential is often assessed based on the volume of buildings to account for multi-floor buildings as well. Consider adding a comment on that in the discussion section.

*(AR): Thank you for this comment. The same approach could be considered in flood risk, especially as stagnant water could still damage property. In this study, however, no buildings were found having more than two levels. A comment of this aspect will be added in the discussion.*

Figure 7 How can the relative error be more than 100%? If the relative error is 100% the objects are not detected at all, right?

*(AR): The relative error is calculated by looking at the absolute difference between a modelled building and a reference building. If this difference is larger than 100% – for example, a 50 $m^2$ modelled and 20 $m^2$ and reference building – this is possible.*

Table 5 It seems like the other approaches may overestimate the damage compared to the UAVbased results. These results are thus more conservative which could be a better estimation in terms of help

for recovery. If the actual damage would be underestimated due to systematic uncertainties handling the UAV-data, this would have consequences for the affected land/home owners. Consider this issue in the discussion section.

*(AR): We thank the reviewer for this comment. What our study suggests is that there is a less uncertainty accompanied with our OBIA approach compared to other approaches. Therefore, we would argue that recovery actions taken on this data would preferable compared to a structural overestimation using a pixel-based approach (which could also yield larger underestimations).*

**Referee #2**

**General**

Based on my field experience, a roof type (in this case, Metal and Thatched) is not enough for a vulnerability classification. What have the authors done to make a further differentiation in terms of building wall material and linking this as the main determinant for damage curve selection?

*(AR): We agree with the reviewer that solely using aerial features as a determinant for damage curve selection has its weaknesses. Here, we used ground observations from our local survey to link the occurrence of wall material to characteristics that could be remotely sensed, being either thatch- or metal-roofing. As mentioned in comments below, we used the percentages in wall-to-roof material to weigh the material-specific damage curves from the CAPRA library. By doing so, we hope to arrive to a damage curve that matches the flood vulnerability of a building more accurately compared to a method which relies only on roof type. Preferably, we would want to know the wall material of each building individually. Therefore, in further studies, we discuss the possibility of using services such as Mapillary or Google Street View to not only derive the distribution of these wall-to-roof relations with greater ease, but also pinpoint the location of these buildings.*

The sensitivity analysis performed does not accommodate a differentiation for building types; only upper and lower damage bounds are used based on curves from metal and thatched buildings respectively. Several studies have shown how damage curves are building-type dependent. Therefore, what is the relevance of the sensitivity analysis?

*(AR): The main purpose of the sensitivity analysis was to highlight the influential parameters in flood damage for hazard, vulnerability, or exposure component. It assesses the effects of the inputs to the uncertainty in the damage outputs. The focus is not on the damage curve per se, but on the different components. This is particularly useful because there is significant uncertainty in each of these them.*

The paper seeks to 'improve flood damage assessment' as captioned in the title. However, the procedure implemented for deducing damage curves (the 'aggregated' curves) for different building types is unclear and should be better addressed

*(AR): We thank the reviewer for this comment. We do think our method still helps to improve flood damage assessments as it provides a better representation of flood damage curves than currently available. Indeed, the damage curves are aggregated but this considers more local data than commonly used in such studies. This aggregation technique is a common approach to aggregate data as also shown for example in Huizenga et al. (2017). This will be better explained in the methods section of the revised manuscript. We will however mention in the discussion other existing methods that could better represent the heterogeneity of depth-damage curves (or damage to buildings curves) for example using Bayesian approaches (Paprotny et al., 2020).*

*Paprotny, D., Kreibich, H., Morales-Nápoles, O., Terefenko, P., and Schröter, K.: Estimating exposure of residential assets to natural hazards in Europe using open data, Nat. Hazards Earth Syst. Sci., 20, 323–343, https://doi.org/10.5194/nhess-20-323-2020, 2020.*

I strongly suggest to separate data and methods. The present form of the manuscript does not clearly separate the two making it difficult to follow. Under the new data section, clearly define the different data layers that are used (satellite data and damage curves), the case study region and also the flood event. these are the core input of the study and deserve separate subsections for clarity.

*(AR): Based on the suggestions of the reviewer, we will separate the data and methods section to improve readability. In addition, we will provide subsections on the case study region and the flood event.*

I will recommend that the authors look at findings from more recent literature on flood damage modelling (or physical vulnerability assessment) in data-scarce areas and discuss similarities or otherwise.

*(AR): We will discuss and add references from more recent literature covering flood damage modelling in data-scare areas, including the paper presented by the reviewer in the comment on line 442, a paper that was briefly mentioned in the introduction by* Englhardt et al., (2019) *and  Rudari et al., (2016).*

*Rudari, Roberto & Beckers, Joost & De Angeli, Silvia & Rossi, Lauro & Trasforini, Eva. (2016). Impact of modelling scale on probabilistic flood risk assessment: the Malawi case. E3S Web of Conferences. 7. 04015. 10.1051/e3sconf/20160704015.*

**Abstract:**

line 17: The phrase 'structural building characteristics' may be misleading. 'Structural' would usually refer to a more complex assessment of the construction technique (e.g. load transfer mechanism, wall compressive/tensile strength, foundation form). For this study, the authors looked at building characteristics (wall material, building size, roof type). Please also adapt in other parts of the manuscript.

*(AR): We agree with the reviewer and 'structural' will be removed or changed to building characteristics.*

**Introduction:**

line 29: Mention specifically how it can support risk reduction

*The sentence will be rephrased to:*

*This can be done a-priori to support strategic risk reduction by, for example, increasing awareness in areas that are high in potential damage and therefore reduce vulnerability, or after a given flood event to quickly derive estimates of building damages to help with recovery and prioritize actions.*

line 35: the sentence says 'much work', therefore you should provide more than one reference

*(AR): Additional references will be added to provide a more convincing argument, including studies by Jongman et al., (2012) and de Moel et al., (2015)*

*Jongman, B., Kreibich, H., Apel, H., Barredo, J. I., Bates, P. D., Feyen, L., Gericke, A., Neal, J., Aerts, J. C. J. H., and Ward, P. J.: Comparative flood damage model assessment: towards a European approach, Nat. Hazards Earth Syst. Sci., 12, 3733–3752, https://doi.org/10.5194/nhess-12-3733-2012, 2012.*

*de Moel, H., Jongman, B., Kreibich, H. et al. Flood risk assessments at different spatial scales. Mitig Adapt Strateg Glob Change 20, 865–890 (2015). https://doi.org/10.1007/s11027-015-9654-z*line 35: Information is available but it is not sufficient. I would the sentence into: Unfortunately, 'sufficient' information…

*(AR): We thank the reviewer for this suggestion. The sentence will be rephased as following:*

*Unfortunately, sufficient information on the exposure and vulnerability is often lacking or less accessible in developing countries.*

line 39: Give example of such studies. Two references should do.

*(AR): In current form, two of such examples are mentioned in the text: Amirebrahimi et al., 2016; Fekete, 2012. In the revised manuscript, this will be clarified.*

line 41: Rephrase sentence for clarity.

*(AR): We rephrased the sentence into:*

*Especially building damage remains hard to quantify, as existing classification categories often neglect spatial heterogeneity.*

line 43-45: Rephrase for clarity

*(AR): The sentence has been rephrased to:*

*Flood damage assessments are a standard procedure to identify economic losses in flood-prone areas. With growing populations and economies, the need to accurately estimate flood damage is gaining greater importance.*

line 66: I would avoid the use of terms like 'accurately' or at least report on model performance and what metric was used to measure it. Admittedly, a lot of progress has been made in flood damage estimation. However, even in data-rich regions, uncertainties still persist in loss prediction.

*(AR): Based on the reviewer suggestion, the performance of the model was added:*

*The model was able to assess damage estimates in an urban setting, with the total average damage deviating from the refund claims with a percentage error lower than 2%.*

line 82: Sentence unclear.

*(AR): The sentence has been rephrased to:*

*By using OBIA for high-resolution imagery, the local spatial heterogeneity between neighboring pixels is conserved instead of leading to an over-classification or salt-and-pepper look in a pixel-based classification (Blaschke, 2010).*

line 96: Please reformulate the sentence for smooth transition. The later part of the introduction focuses more on remote sensing techniques. The sentence 'From the above it is clear that exposure and vulnerability components are underrepresented' does not simply fit.

*(AR): To facilitate a smooth transition the authors have decided to delete the sentence, starting the paragraph with.: "In this research, we aim to bridge the gap...". In addition, we have shortened the previous paragraph to improve coherence of the introduction.*

**Data and Methods:**

The information provided in lines 108 – 119 can be streamlined. It contains some general information that should be avoided in a data/methods section.

lines 121 – 125: Repetition

*(AR): As suggested, the section will be revised to improve structure and avoid repetition of information.*

line 123: The paper title reads 'improvement of damage assessment'. Therefore, the stage- damage curves used is a key component on which the results of this studies relies on. You have to clearly outline what data was used and how it was generated. If it is from another study, it should be properly cited and at least briefly mention the input data

*(AR): To provide the rationale behind our generated damage curves, an aggregation table on the material-specific damage curves from the CAPRA library will be added. In addition, it will be specified how the damage curves were created using the information on local building stock. The comment on line 315 goes into more detail about this process.*

line 80: …building elements? Please be consistent with the term.

line 80: Not an information for data/methods. Can be deleted

*(AR): we are not entirely clear what section the reviewer refers to here (line 80 is in the introduction). Could the reviewer please clarify?*

line 137 – 140: Provide a reference on the responsible agency that compiled the data on people affected and damages. (this cannot be the volunteers - correct me if i am wrong). And the term 'volunteers' should be more clearly defined. Are they community residents? Workers from the red cross? please specify.

*(AR): The information on affected people and damages, derived from the EPoA of the IFRC (2019), was based on an assessment carried out by the Village Civil Protection Committee (VCPC) and Malawi Red Cross Society (MRCS) volunteers from the District branch office. Although it is not specifically mentioned, this is often a combination of staff members from the Red Cross and (local) volunteers from the district. The MRCS counts 163 members of staff, 76,000 volunteers and 33 branches across the country (IFRC, 2019). It will be clarified in the manuscript that this data was compiled by the IFRC (2019).*

Also, where there no local media that reported the event, number of affected persons or incurred damages?

*(AR): To the best of our knowledge, no local damage reports of the villages covered in our case study area were published or mentioned. The most detailed information covering our study area is aggregated on district level (TA Makhuwira) in the EpoA by the Red Cross, which, unfortunately, complicates validating our results with observed damages.*

Figure 1: Under vulnerability, damage curves are used which normally combines data on hazard and building type. You have to define what you refer to as 'vulnerability'

*(AR): We thank the reviewer for this comment. We agree that vulnerability has a broad meaning, including for example susceptibility, resilience, or coping capacity. The components of hazard, exposure and vulnerability are briefly mentioned in the introduction: "Flood risk is defined as a combination of the elements: hazard (flood extent and depth), exposure (exposed assets) and the conditions of vulnerability that are present (the susceptibility of buildings to floods) (UNDRR, 2019)". We thus use vulnerability as the structural vulnerability of buildings to flooding. In the figure, these definitions will be added to clarify the terms that are used.*

line 169: What is the total number of buildings in this region? What kind of sampling technique was implemented (random or systematic) and why? Also, why are 50 buildings considered representative of building types?

*(AR): For the three villages in the case study, the OSM layer contains 1348 buildings. The OBIA method detected 1466 buildings. Although these numbers seem to correspond, the statistical accuracy of the OBIA should be considered. This means that the OBIA dataset also contains several False Positives and Negatives.*

*The sample buildings were collected by a Red Cross volunteer / MSc. student that conducted several interviews in the case study area for a thesis research on early warning systems for floods. The data was collected by observing buildings around the sites of these interviews. Ideally, a larger sample size would have been acquired to provide a statistically sound justification for using this data to create damage curves. Unfortunately, a narrow time window and limited means of communication and accessibility meant that this sample size was all that could be achieved. The results of this study point to a compelling need to test the method both with a larger sample and with a variety of sampling techniques but does provide a framework for similar studies in the future.*

line 171: why are these considered as flood vulnerability parameters? where their selection based on previous studies, expert-knowledge, author knowledge or due to their simplicity for integration into a satellite-based rapid vulnerability assessment? Please clarify these.

*(AR): The flood vulnerability parameters were selected based on key features that characterize building types in the region as described in the national Intergrated Household Survey, in addition to being relatively easy to detect from remote imagery. This will be clarifed in the revised manuscript.*

Figure 2: Include the North arrow.

*(AR): A North arrow will be included in the figure.*

line 190: Segmentation 5m2 . Typo?

*(AR): The third step of the segmentation process is setting threshold for region size. This way only meaningful regions are created. The sentence will be rephrased to:*

*… the minimum size of a segmented region to 5m², in relation to minimum building sizes.*

line 302: Be more explicit and avoid vague comments. State what building typologies are within the area and state the type of buildings that the damage curves (used in your study) were aggregated from.

*(AR): we agree with the reviewer and in the revised manuscript we will mention which building typologies are in the area and state the type of buildings for the damage curves. For the district of Chikwawa, the Integrated Household Survey reports a distribution in traditional, semi-permanent and permanent buildings of 32,5%, 33,8% and 33,7%, respectively. Examples of these buildings will be added to the supplementary data.*

line 304: A sentence or two on the method used in the CAPRA library and input data should be provided. What information specifically was deduced from Maiti (2007) for the adjustments. Be specific.

*(AR): As suggested by the reviewer, in the revised manuscript we will provide more details on the method of creating damage curves using the CAPRA library and Maiti (2007). See the comment on line 123.*

line 315: These type of sentence should be avoided. Properly communicate what was done in this step. What is the local distribution of the regional building type? How where the curves aggregated?

*(AR): For the district of Chikwawa, the Integrated Household Survey reports a distribution in traditional, semi-permanent and permanent buildings of 32,5%, 33,8% and 33,7%, respectively. These percentages are used to calculate the damage corresponding to one unit in the pixel-based approach. The damage curves are constructed by combining the material-specific curves from the CAPRA library based on the building description from the Integrated Household Survey. For the object-based approach, our local survey data is used to construct two building types based on their distinctive aerial features, being metal and thatch. For the metal-roofed buildings, two wall materials were found: burnt red bricks (90%) and unburnt bricks (10%). For the thatch-roofed buildings, three wall materials were found: burnt red bricks (27%), unburnt bricks (41%) and mud/wattle (32%). These percentages were used to weigh the material-specific curves from the CAPRA library and create a damage curve for both metal and thatch buildings. The percentages and aggregation technique will be added in more detail to the manuscript to clarify the method.*

**Results:**

line 340: Based on my field experience, a roof type (in this case, Metal and Thatched) is not enough for a vulnerability classification. For example, buildings with Thatched roofs can have either a mud or unburnt bricks, or even burnt bricks as wall material, which in this case plays a more important role for flood vulnerability. What have the authors done to make a further differentiation in terms of building wall material and linking this as the main determinant for damage curve selection?

*(AR): We agree that solely using aerial features as a determinant for damage curve selection has its weaknesses. Here, we used ground observations from our local survey to link the occurrence of wall material to characteristics that could be remotely sensed, being either thatch- or metal-roofing. As mentioned in a comment above, we used the percentages in wall-to-roof material to weigh the material-specific damage curves from the CAPRA library. By doing so, we hope to arrive to a damage curve that matches the flood vulnerability of a building more accurately compared to a method which relies only on roof type. Preferably, we would want to know the wall material of each building individually. Therefore, in further studies, we discuss the possibility of using services like Mapillary or Google Street View to not only derive the distribution of these wall-to-roof relations with greater ease and in larger sample size, but also pinpoint the location of these walls and combine them with remotely sensed data.*

line 367: Copier ref…Typo?

*(AR): A reference should be mentioned here. The typo will be removed in our revised version.*

line 360: The difference is substantial enough to alter the actual damage especially given that stage-damage curves primarily depend on water depth at building locations. What is your take on this?

*(AR): We agree that this difference is substantial and believe that this aspect deserves attention in further research, especially given the tendency of certain buildings in the area to collapse at relatively low inundation levels.*

*Using the surface water interpolation, we found considerable differences with the hydraulic model along the main channel. This is in line with research from Cohen et al., (2018), given the inability of similar methods to calculate fluid dynamic effects. Most buildings, however, are in the floodplain, further away from the main channel where large differences between this method and the hydraulic model occur. This means that a larger RMSE, caused by these differences, is not necessarily a reason to*

*abandon our proposed approach. We do think that using the surface water interpolation method is a computationally simple method, using only imagery of a floodplain and a DEM. In addition, it does not rely on observed discharge values, contrary to hydraulic modelling, which makes it functional in data-scarce areas.*

*Furthermore, it should be noted that both the hydraulic simulation and the surface water interpolation method have their disadvantages. In the hydraulic simulation, for instance, we lacked specific discharge information for the event so an estimate had to be made there that would resemble flood levels. The DEM method, on the other hand, lacks the dynamics of the hydraulic simulation, but is more specific to the event considered in this study as it is based on the observed extent.*

*We decided not to further focus on this aspect in the paper, as the scope of this research was on the added value of using high-resolution imagery in detecting building objects and the comparison between an individual object approach and an aggregated land-use approach. In both approaches, the same water depth is used to compute flood damage. Therefore, we would argue that the exposure and vulnerability estimation is of greater importance to our objective than the water depth estimation.*

line 360: Since the flood occurred in 2019, it is likely that the residents of the study area will remember very closely the flood depth in their houses. Did the authors attempt to interview the residents, in this way integrating community-based approaches (e.g. citizen science). Such data extraction methods are becoming more popular in data-scarce regions.

*(AR): The reviewer has a very good point. This would have been a very nice addition to our research, especially for validating the surface interpolation method. At the moment of collecting field data, we did not take into account certain methods and no interviews were conducted on historic flood depths. Further research that includes community-based approaches could provide the evidence necessary for confidently accepting the surface water interpolation method. This will be added to the discussion.*

line 362: What is the RMSE between the observed and modelled flood depth?

*(AR): Unfortunately, no observed water depth values were collected in the flooded area. Therefore, a tributary of the river, for which high-resolution imagery was available, was taken to validate the results of the of the surface water interpolation against a hydraulic model run. Subsequently, the surface water interpolation technique was used on the entire case study area. For most of this area, only a SRTM DEM (30m) was available. Although the discharge values used in this model could not be validated against the actual flood event, it does provide an estimation of potential water depth and extent associated with flooding in the area.*

line 423: Why is this the first time the ERN data base is mentioned? Did I miss the earlier reference…

*(AR): CAPRA is the platform responsible for the ERN data base. In the revised manuscript, this will be changed to 'CAPRA library' to avoid confusion.*

line 426: The sensitivity analysis does not accommodate a differentiation for building types; only upper and lower damage bounds are used based on curves from metal and thatched buildings, respectively. Several studies have shown how damage curves are building-type dependent. Therefore, what is the relevance of the sensitivity analysis? What does it really communicate in terms of damage uncertainty for regional building types. I doubt if this sensitivity approach is ideal for the case study region considering data scarcity.

*(AR): The main purpose of the sensitivity analysis was to highlight the influential parameters in flood damage for hazard, vulnerability, or exposure component. It assesses the effects of the inputs to the*

*uncertainty in the damage outputs. The focus is not on the damage curve per se, but on the different components. This is particularly useful because there is significant uncertainty in each of these them.*

line 431: There are more recent studies discussing this… One or two references will do

*(AR): the revised manuscript will include references to some more recent studies,*

line 433: Check more recent studies

*(AR): Based on the reviewers' suggestion, more recent references on the relative contribution of each model component to the uncertainty in flood loss estimation will be added and discussed, including a study by Winter et al., (2018) and Saint-Geours et al ., (2015).*

*Winter, B., Schneeberger, K., Huttenlau, M. et al. Sources of uncertainty in a probabilistic flood risk model. Nat Hazards 91, 431–446 (2018). https://doi.org/10.1007/s11069-017-3135-5*

*Nathalie Saint-Geours, Frédéric Grelot, Jean-Stéphane Bailly, Christian Lavergne. Ranking sources of uncertainty in flood damage modelling: a case study on the cost-benefit analysis of a flood mitigation project in the Orb Delta, France. Journal of Flood Risk Management, Wiley, 2015, 8 (2), pp.161-176.  ⟨10.1111/jfr3.12068⟩. ⟨hal-00762009⟩*

**Discussion:**

line 442: 'uncovered several important factors in ...flood damage estimation'. Maybe I missed this but the current structure of the manuscript makes it difficult to assess the contributions or novel aspects. See also a recent research on flood damage modelling in data-scarce areas dealing with comparable building types and discuss relevant similar findings or otherwise (Malgwi MB, Schlögl M, Keiler M (2021): Expert-based versus datadriven flood damage models: a comparative evaluation for data-scarce regions (https://doi.org/10.1016/j.ijdrr.2021.102148))

*(AR): Thank you for presenting this study. The article discusses a very interesting comparison between an expert-based model and multivariate model, both focusing on physical vulnerability. Several similarities can be found with our research. An interesting similarity that the authors touch upon is the uncertainty associated with expert judgement. They state that expert interviews are generally subjective and that getting required 'expertise' is not always feasible. If that is the case, this expertise will have to be transferred from other regions. This expertise – or rather the absence of it – can be compared to a pixel-based approach. Here, information of a certain region is projected on a local scale or on another region. The main insight that our study provides, is that OBIA has the potential to objectively classify buildings on a local scale, therefore improving on a pixel-based flood damage assessment that uses downscaled information from a larger level. This is because pixel-based information is inherently less representative for individual buildings considering the spatial heterogeneity of building types.*

*Another aspect that is discussed by the authors is the field data collection. Basic input for their models is the field data collected by interviews, which means that the output also relies on this information and a large sample size is required. Especially the scarcity of empirical loss data hinders the development and application of multivariate models in developing countries. What our study indicates, it that OBIA combined with local data – preferably already existing street view imagery – can accurately estimate flood damage in an area where such data is absent, for example due to its remoteness. Obviously, collecting UAV data and local building information requires a substantial effort as well. A promising line of research would be to compare our improved object-based approach using OBIA with*

*a data-driven multivariate approach, both in result and required effort. This study and its findings will be discussed in the revised manuscript.*

line 480: typo

*(AR): The duplicate will be removed.*

---

## Referee Report (RR1)

Referee 2_Report

The manuscript has substantially improved from the first version. I commend the authors for their detailed response to all concerns raised during the review. I recommend the manuscript to be accepted as it is.

For future purpose, I recommend that the author(s) include line numbers in their response to reviewer comments so that reviewers can easily (and in short time) navigate to find where specific comments have been addressed.

---

## Author Response (AR2)

**Authors' reply to referees' comments**

We would like to thank the reviewer for the constructive comments and the time spent on the review of this article. Following the suggestions of the reviewer, the manuscript has been edited at several places:

- Several textual changes have been made which are specified in the 'Tracked Changes' document. In addition, the figures were edited.
- The Study Area subsection is now an individual section.
- Based on the first revision of the manuscript, the former "Data & Methods" was renamed "Material & Methods". However, as strongly suggested by referee #2, we separated the sections to improve readability.
- An alternative method and reference on automatized filtering to compute a DTM has been added to the discussion.
- Based on the suggestion of the referee to validate the surface water interpolation, ROC curves were plotted and the AUC calculated. As the flood map for the total case study area is derived from SRTM DEM and SAR imagery, and there is no ground truth map to create thresholds for an ROC analysis, we mention the need of validating our proposed method using global DEM's (such as SRTM) in future studies.
- A relevant paper on physical vulnerability assessment in data-scare areas was added to the introduction.

We hope that the revised manuscript will meet your expectations after processing the minor comments. Thank you for your time and efforts, and for considering our manuscript for a possible publication in NHESS.

Yours Sincerely,

Lucas Wouters, on behalf of all co-authors.